# Gut microbiome composition and metabolic activity in women with diverticulitis

Wenjie Ma[1,2], Yiqing Wang[1,2], Long H. Nguyen [1,2], Raaj S. Mehta[1,2], Jane Ha[1,2], Amrisha Bhosle[3,4], Lauren J. McIver[3], Mingyang Song[1,2,5,6], Clary B. Clish [4], Lisa L. Strate[7,11], Curtis Huttenhower [3,4,11] & Andrew T. Chan [1,2,4,8,9,10,11] ✉

The etiopathogenesis of diverticulitis, among the most common gastrointestinal diagnoses, remains largely unknown. By leveraging stool collected within a large prospective cohort, we performed shotgun metagenomic sequencing and untargeted metabolomics profiling among 121 women diagnosed with diverticulitis requiring antibiotics or hospitalizations (cases), matched to 121 women without diverticulitis (controls) according to age and race. Overall microbial community structure and metabolomic profiles differed in diverticulitis cases compared to controls, including enrichment of pro-inflammatory *Ruminococcus gnavus*, 1,7-dimethyluric acid, and histidine-related metabolites, and depletion of butyrate-producing bacteria and anti-inflammatory ceramides. Through integrated multi-omic analysis, we detected covarying microbial and metabolic features, such as *Bilophila wadsworthia* and bile acids, specific to diverticulitis. Additionally, we observed that microbial composition modulated the protective association between a prudent fiber-rich diet and diverticulitis. Our findings offer insights into the perturbations in inflammation-related microbial and metabolic signatures associated with diverticulitis, supporting the potential of microbial-based diagnostics and therapeutic targets.

Two-thirds of U.S. adults aged 70 years or older have diverticulosis[1], or the presence of largely asymptomatic and often innumerable small pouches in the colon. However, this condition may predispose individuals to acute diverticulitis–or inflammation of one or more of these pouches–which can then culminate in other complications, including obstruction, fistula or abscess formation, or peritonitis[2]. More than 20% of patients will experience a recurrent episode[3]. Despite the recent statistics showing that diverticulitis is among the

top common gastrointestinal disorders responsible for millions of healthcare encounters[4], the pathogenesis of diverticulitis remains largely unclear.

Perturbations in the gut microbiota have been recognized as a critical etiological factor for inflammatory conditions of the colon, such as inflammatory bowel disease (IBD)[5,6] and disorders of the gut-brain axis, including irritable bowel syndrome[7]. Diverticulitis shares symptoms, histological features, and inflammatory changes with these

¹Clinical and Translational Epidemiology Unit, Massachusetts General Hospital and Harvard Medical School, Boston, MA, USA. ²Division of Gastroenterology, Massachusetts General Hospital and Harvard Medical School, Boston, MA, USA. ³Department of Biostatistics, Harvard T.H. Chan School of Public Health, Boston, MA, USA. ⁴Broad Institute of MIT and Harvard, Cambridge, MA, USA. ⁵Department of Epidemiology, Harvard T.H. Chan School of Public Health, Boston, MA, USA. ⁶Department of Nutrition, Harvard T.H. Chan School of Public Health, Boston, MA, USA. ⁷Division of Gastroenterology, University of Washington School of Medicine, Seattle, WA, USA. ⁸Department of Immunology and Infectious Diseases, Harvard T.H. Chan School of Public Health, Boston, MA, USA. ⁹Cancer Center, Massachusetts General Hospital, Boston, MA, USA. ¹⁰Channing Division of Network Medicine, Department of Medicine, Brigham and Women's Hospital and Harvard Medical School, Boston, MA, USA. ¹¹These authors contributed equally: Lisa L. Strate, Curtis Huttenhower, Andrew T. Chan. ✉e-mail: achan@mgh.harvard.edu

two diseases[8–10]. The recent theory of diverticulitis pathogenesis supports a role of gut microbiota in the pathogenesis of this disease as well[11]. Acute diverticulitis involves micro- or macro-perforations with translocation of bacteria across the colon mucosal barrier, which sometimes results in systemic inflammation and complications. In such a setting, antibiotics are the first-line therapy[12] though expectant management is acceptable in some cases. Lifestyle and dietary factors that have been consistently linked to diverticulitis, such as obesity[13,14] and a Western diet including low fiber intake[15–17], have an impact on the composition and function of the gut microbiome, including altered metabolism of short-chain fatty acids and bile acids with attendant intestinal barrier dysfunction, leading to mucosal inflammation and subsequent development of diverticulitis[18,19].

Several limited studies have shown shifts in the gut microbial composition of patients with diverticulitis[20,21]. Patients with diverticulitis had higher levels of *Bifidobacterium longum* than those with colon cancer or IBD in surgically resected mucosal specimens using a quantitative polymerase chain reaction (qPCR) analysis[20]. Daniels et al. showed that patients with uncomplicated diverticulitis had a higher diversity of Proteobacteria, mostly Enterobacteriaceae, in the fecal microbiota compared to control patients with mixed gastrointestinal conditions[21]. Other studies have focused on the microbiota of symptomatic uncomplicated diverticular disease (SUDD) without diverticulitis[22–24]. For example, in fecal samples among patients with SUDD, there were depleted levels of microbe members with anti-inflammatory properties such as *Clostridium cluster IV*, *Clostridium cluster IX*, *Fusobacterium*, and Lactobacillaceae[23].

However, each of these prior investigations of gut microbiota and diverticulitis has relied on small cohorts or amplicon-based characterizations of the microbial community. Whole genome shotgun metagenomics offers finer resolution for microbial community structure and dynamics compared to 16 S sequencing. In this work, we performed the largest shotgun metagenomic sequencing study on the topic to-date, coupled with untargeted LC-MS metabolomic profiling of stool samples from 242 patients with diverticulitis and matched, non-diverticulitis controls. We successfully identified distinct microbial species, enzymes, and metabolites that undergo specific changes in diverticulitis, as well as any co-occurring microbe-metabolite associations relevant to the disease.

## Results

We conducted a nested case-control study among 14,992 women from the Nurses' Health Study II who returned a stool sample from February 2019 to July 2021 as part of the large-scale Microbiome among Nurses (Micro-N) study (Methods, Fig. 1a)[25]. We identified 121 incident cases of diverticulitis requiring antibiotics or hospitalizations based on self-report from the 2019 biennial cohort questionnaire and matched them to 121 diverticulitis-free controls according to age, race, and month of stool collection (total N = 242). We performed shotgun metagenomic sequencing that was subsequently profiled to quantify microbial taxa, genes, and pathways, plus untargeted metabolomics profiling for metabolites including lipids, polar metabolites, free fatty acids, and bile acids (Methods). As expected, women with diverticulitis had higher body mass index (BMI) and lower physical activity, fiber intake,

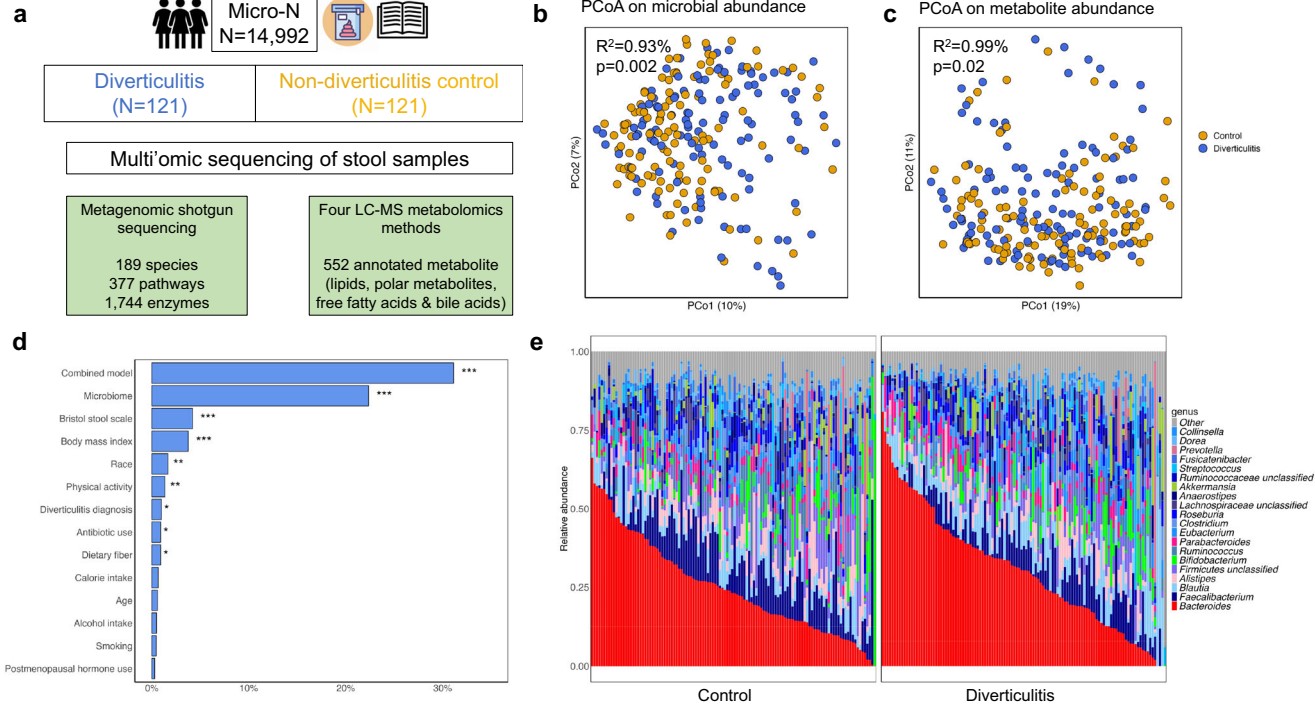

**Fig. 1 | Alterations of gut microbial community structure in diverticulitis.**
**a** Among 14,992 women from the Micro-N study who returned a stool sample from February 2019 to July 2021, we identified 121 women with diverticulitis and matched them to 121 diverticulitis-free controls according to age, race, and month of stool collection. We performed whole genome shotgun sequencing and untargeted metabolomics profiling of the stool samples. **b** PERMANOVA of Bray-Curtis dissimilarities of species suggested significant differences in the overall microbial community in diverticulitis and controls (two-sided, unadjusted *p* = 0.002). Data from 117 diverticulitis cases and 118 controls were included in the microbiome analysis. **c** PERMANOVA of Bray-Curtis dissimilarities of metabolomics profile (552 known metabolites) suggested significant differences in the overall metabolomic

composition in diverticulitis and controls (two-sided, unadjusted *p* = 0.02). Data from 116 diverticulitis cases and 116 controls were included in the metabolomic analysis. **d** PERMANOVA of Bray–Curtis dissimilarities of metabolomics profile explained by the microbiome, diverticulitis, and other factors. *P* values are two-sided, with multiple comparison corrected using Benjamini-Hochberg false discovery rate (FDR). Data from 113 diverticulitis cases and 113 controls were included in the microbiome-metabolite paired analysis. *** FDR *p* < 0.01; **0.01 < FDR *p* < 0.05; *0.05 < FDR *p* < 0.1. Exact R² and *p* values are included in Supplementary Data 1. **e** Differences in relative abundances of the top 20 genera, including enriched *Bacteroides* but depleted *Faecalibacterium* in diverticulitis.

**Table 1 | Characteristics of diverticulitis cases and diverticulitis-free controls**

| | Diverticulitis (n = 121) | Control (n = 121) |
|---|---|---|
| Age | 64.9 (4.7) | 64.7 (4.6) |
| White | 98.3 | 98.3 |
| Body mass index | 28.1 (6) | 27.2 (6.4) |
| Total physical activity (MET-h/week) | 26.0 (26.6) | 29.6 (30.3) |
| Current smoker, % | 0.8 | 2.5 |
| Menopausal hormone use, % | 22.3 | 19.8 |
| Antibiotic use in the past month, % | 12.4 | 7.4 |
| Total fiber intake, g/d | 23.0 (6.6) | 24.5 (5.7) |
| Total calorie intake, kcal/d | 1860 (501) | 1859 (560) |
| Alcohol intake, g/d | 7.6 (8.7) | 7.4 (9.6) |
| Alternate Healthy Eating Index-2010 | 62.0 (12.6) | 64.5 (11.3) |
| Prudent dietary score | 0.1 (0.9) | 0.3 (0.9) |
| Western dietary score | 0 (0.9) | −0.1 (0.8) |
| Bristol stool scale[a], % | | |
| Hard | 12.5 | 12.4 |
| Normal | 62.5 | 69.4 |
| Soft | 25.0 | 18.2 |

Cases and controls were matched according to age, race, and month of stool collection. Values are means (SD) or percentages.
[a]Bristol stool scale was assessed for the collected stool sample.

and overall dietary quality (Table 1). A higher percentage of women with diverticulitis reported antibiotic use during the past year compared to those without diverticulitis.

**Microbial composition and functions in diverticulitis**

Overall microbial community structure (Fig. 1b) and activity (Fig. 1c) differed slightly in diverticulitis and controls. Using omnibus testing with PERMANOVA of Bray-Curtis dissimilarities, diverticulitis explained a small but significant amount of the variation of the profiles of the microbiome ($R^2 = 0.93\%$; $p = 0.002$) and metabolomics ($R^2 = 0.99\%$; $p = 0.02$). The gut microbiome was the leading factor explaining the variance in the metabolomics profile ($R^2 = 22.4\%$), followed by the Bristol stool scale ($R^2 = 4.19\%$) and BMI ($R^2 = 3.76\%$; Fig. 1d, Supplementary Data 1). Patients with diverticulitis had a significant reduction in species richness ($p = 0.024$) and a non-significant reduction in the Shannon index (Supplementary Fig. 1). Among the top genera, there were enrichments of *Bacteroides* and *Blautia* but depletions of *Faecalibacterium* and unclassified Firmicutes members in diverticulitis (Fig. 1e).

A total of 189 microbial species remained in the analysis with the minimum prevalence (10% of samples) and relative abundance (0.01%) threshold. We identified individual species altered in diverticulitis using a multivariate linear model in MaAsLin 2 (Fig. 2, Supplementary Data 2). Several *Clostridium* species were enriched in diverticulitis, whereas *Eubacterium* and *Faecalibacterium* species were depleted. Some of these associations remained despite additional adjustments of lifestyle and dietary variables, including increases of the *Clostridium* species, *Erysipelatoclostridium ramosum*, and *Hungatella hathewayi* that can be opportunistic pathogens[26,27]; increases of *Blautia* and *Ruminococcus gnavus* that also found to be enriched in IBD[28,29]; decreases of butyrate producers such as *Eubacterium eligens*, *Faecalibacterium*, and *Subdoligranulum sp APC924 74*[30,31]; as well as *Parasutterella excrementihominis*, which is involved in lipid metabolism[32,33]. In particular, among the species that were significantly altered in diverticulitis, *Anaerotruncus colihominis*, a relatively newly described

species involved in sulfur metabolism[34,35] that has been positively correlated with bacteremia[36] and multiple sclerosis[37], was found to be increased in severe diverticulitis cases that required surgery or had an abscess compared to non-severe cases (Supplementary Fig. 2, Supplementary Data 3). These taxonomic changes corresponded to those observed in IBD[38] (Supplementary Fig. 3, Supplementary Data 4).

We built a Random Forest classifier based on the relative abundances of all 189 retained species to discriminate between diverticulitis and controls, which yielded a mean area under the Receiver Operating Characteristics curve (AUC) of 0.66 (±0.06) over 100 iterations (Supplementary Fig. 4). This was slightly lower than the discriminatory power of taxonomic composition in other conditions such as colorectal cancer[39] or IBD[40]. Consistent with results from multivariate testing above, *P. excrementihominis* showed the greatest discriminative value, followed by *Clostridium* species, *E. eligens*, *H. hathewayi*, and *R. gnavus*. Additional incorporation of annotated metabolites or lifestyle factors such as body mass index and fiber intake did not improve the AUC.

Unlike the major shifts in the overall structure and taxonomic composition, we only observed modest changes in microbial pathways during multivariate testing (Supplementary Data 5). In the fully adjusted model, there was a statistically significant enrichment of the pathway involved in succinate fermentation to butanoate (PWY-5677) in diverticulitis (Fig. 3a), which was contributed solely by increases in *Flavonifractor plautii*. Enrichment of phosphatidylglycerol biosynthesis II (PWY4FS-8) was also observed (Fig. 3b), contributed primarily by *Fusicatenibacter saccharivorans*, *Escherichia coli*, and *R. gnavus*. Among 1744 microbial enzymes that passed filtering, 110 were significantly altered in diverticulitis (Supplementary Data 6). While many of these comprise cellular housekeeping genes, some have more tailored functions. For example, consistent with the finding of enriched fatty acid biosynthesis, we found increases in acetyl-CoA carboxylase (EC 6.4.1.2) and the component-biotin carboxylase (EC 6.3.4.14), key enzymes catalyzing the transfer of a carboxyl group from hydrogencarbonate to protein-carried biotin and forming malonyl-CoA in the de novo fatty acid biosynthesis (Fig. 3c). Significant increases were also seen on diaminopimelate dehydrogenase (EC 1.4.1.16), involved in the biosynthesis of lysine, and aspartate transaminase (EC 2.6.1.1) that plays a role in the metabolism of arginine biosynthesis. Finally, reductions of some enzymes were detected in diverticulitis compared to controls, such as citrate (pro-3S)-lyase (EC 4.1.3.6), forming acetate from citrate, and glutamate-tRNA ligase (EC 6.1.1.17).

**Gut microbiome composition modulates the protective effects of fiber on diverticulitis**

Consistent with our previous findings as well as prior epidemiological studies[16,17,41], we found greater dietary fiber intake was associated with a lower risk of diverticulitis. Compared to women in the lowest quartile of fiber intake, those in the highest quartile had an odds ratio of 0.36 in developing diverticulitis (95% CI, 0.16–0.83). Adherence to a prudent diet showed a similar trend, which was largely dependent on fiber intake, whereas adherence to a Western diet was not significantly associated (Supplementary Data 7).

Interindividual variation in the gut microbiome composition has been linked to the differential effects of diet on chronic inflammation and various disease outcomes[42–44]. Here, adherence to a prudent diet and greater fiber intake was associated with a greater reduction in the risk of diverticulitis in individuals with microbial configurations corresponding with high loadings on the second ordination principal coordinate (PCo2, Fig. 4a/b, Supplementary Data 8). *Bacteroides uniformis* was the most abundant species and the top determinant of PCo2 in our samples, although this also corresponded with high loadings of other *Bacteroides*/Bacteroidetes members. Consistently, a prudent diet was associated with a greater risk reduction in individuals with high *B. uniformis* (Fig. 4c). *B. uniformis* and other *Bacteroides* have

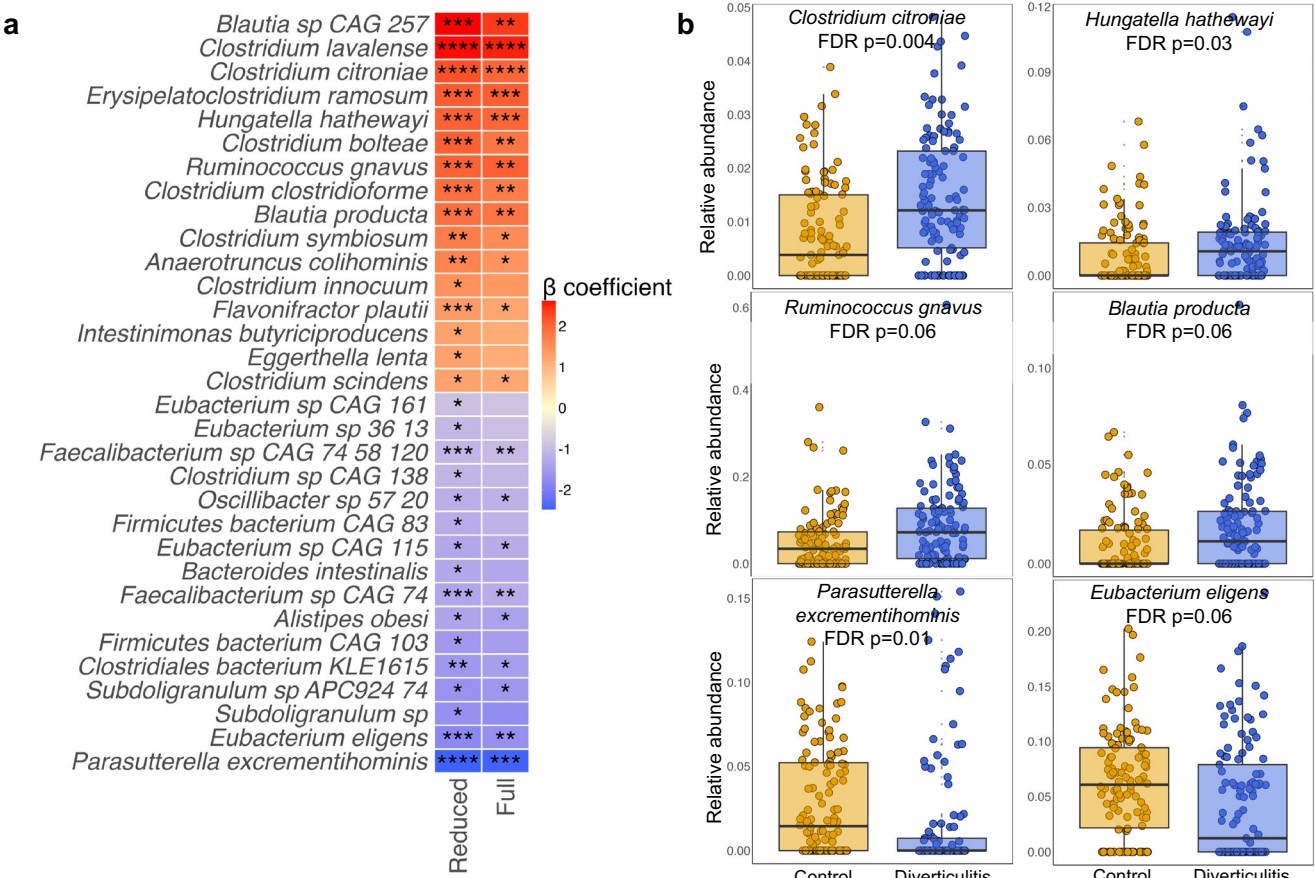

**Fig. 2 | Species abundances statistically significantly altered in diverticulitis. a** β coefficient of the statistically significant associations between species and diverticulitis. We adjusted for age, race, Bristol stool scale, and antibiotics use in the reduced model, and the full model was further adjusted for fiber intake, alcohol consumption, body mass index, smoking, menopausal hormone use, physical activity, Alternate Healthy Eating Index, and calorie intake. *P* values are two-sided, with multiple comparison corrected using Benjamini-Hochberg false discovery rate (FDR). **** FDR *p* < 0.01; *** 0.01 < FDR *p* < 0.05; ** 0.05 < FDR *p* < 0.1; * 0.1 < FDR *p* < 0.25. Complete results for associations between species and diverticulitis from MaAsLin 2 models are provided in Supplementary Data 2. **b** Examples of arcsin square root transformed abundances of species altered in diverticulitis. Data are presented as median with the lower and upper hinges corresponding to the 25% and 75% percentiles. The lower and upper whiskers show the smallest or largest value within 1.5 IQR (inter-quartile range) from the nearest hinge.

been shown to induce distinct cytokine responses and influence obesity-related inflammation[45,46], and *B. uniformis* combined with fiber led to the greatest metabolic and immune benefits in obese mice compared with separate administration[47].

Relatedly, *Prevotella copri* has also been consistently shown to be a unique interactor with dietary effects on host metabolic and inflammatory parameters[42,48,49]. While the number of participants who carried *P. copri* in our population was limited (n = 29), we did observe a similar trend that having *P. copri* mitigated the protective association between fiber and diverticulitis (Supplementary Fig. 5).

### Metabolomic shifts in diverticulitis include enrichments of metabolites related to histidine metabolism and depletions of microbially associated ceramides

To complement our microbial community profiling of the gut in diverticulitis, we also assayed untargeted small molecules in all samples. The total number of stool metabolomic features detected after quality control was similar in diverticulitis and controls (*N* = 51,048 and 50,481, *p* = 0.42; Methods). These included 523 annotated metabolites belonging to 84 chemical subclasses. Upon comparing their mean abundances between diverticulitis and controls (Fig. 5, Supplementary Data 9), we observed that patients with diverticulitis had enriched organosulfonic acids and derivatives, carboxylic acid derivatives, and indolyl carboxylic acids and derivatives, as well as depleted diterpenoids, ceramides, pyrrolidinylpyridines, indolines, fatty alcohols, and

bilirubins. Several of these metabolite groups, such as ceramides, a subgroup of sphingolipids, can be produced by *Bacteroides*[50] and have been implicated in inflammation pathways[51], IBD[5], and cardiometabolic diseases[52].

Individual metabolites from most of these classes were also significantly associated with diverticulitis in the multivariate model, such as enriched metabolites related to histidine metabolism including 3-methylhistidine and N-acetylhistamine, docosapentaenoic acid (DPA), 1,7-dimethyluric acid, and PGF2beta, as well as reduced ceramides, 3-methyladipic acid, N-acetylglutamic acid, isoallolithocholic acid, LPE 18:1, and stearic acid (Fig. 6, Supplementary Data 10). Some metabolites were related to the severity of diverticulitis (Supplementary Fig. 6, Supplementary Data 11). For example, 1-methyluric acid was found to be increased, whereas 5-hydroxy-tryptophan was reduced in severe diverticulitis. These results also support similarities of metabolomic perturbations in IBD. For example, increased histidine metabolism-related metabolites and reduced L-urobilin were shown as the most important features distinguishing dextran sulfate sodium(DSS)-induced colitis[53]. Similarly, DPA, one of the n-3 polyunsaturated fatty acids, was significantly elevated in patients with Crohn's disease, possibly due to a reduced absorption and corresponding reduced anti-inflammatory effect in the disease[54].

We further included the much larger group of unannotated metabolites (*n* = 56,937 in total) using MACARRoN (https://huttenhower.sph.harvard.edu/macarron/), a computational workflow

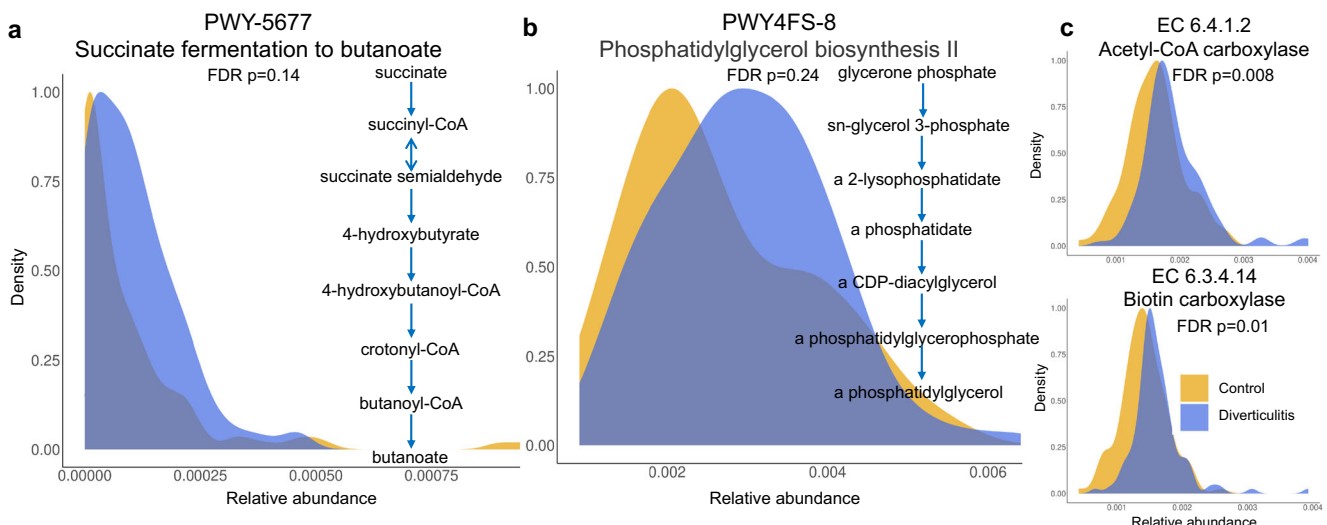

**Fig. 3 | Metagenomic pathways and functional potential related to fatty acid and short-chain fatty acids statistically significantly altered in diverticulitis.** We adjusted for age, race, Bristol stool scale, and antibiotics use in the reduced model, and the full model was further adjusted for fiber intake, alcohol consumption, body mass index, smoking, menopausal hormone use, physical activity, Alternate Healthy Eating Index, and calorie intake. P-values are two-sided, with multiple comparison corrected using Benjamini-Hochberg false discovery rate (FDR). Complete results for associations of all pathways and ECs with diverticulitis from MaAsLin 2 models are provided in Supplementary Data 5 and 6. **a** PWY-5677: succinate fermentation to butanoate enriched in diverticulitis. **b** PWY4FS-8: phosphatidylglycerol biosynthesis II enriched in diverticulitis. **c** Acetyl-CoA carboxylase (EC 6.4.1.2) and the component-biotin carboxylase (EC 6.3.4.14), two enzymes involved in the de novo fatty acid biosynthesis enriched in diverticulitis.

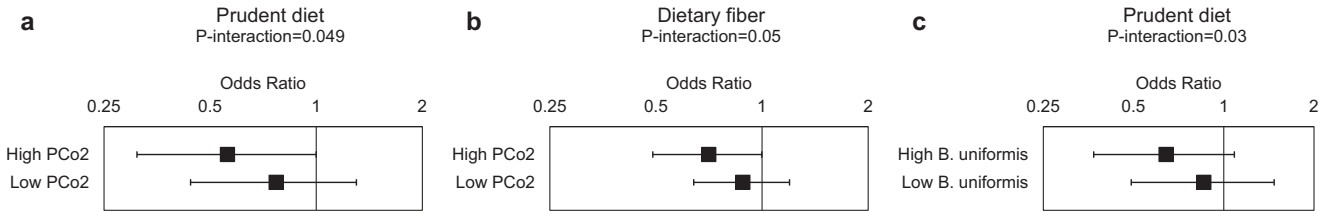

**Fig. 4 | Gut microbiome composition modulates the association between a prudent diet and diverticulitis.** We evaluated the associations of a prudent diet and fiber intake with the risk of diverticulitis in subgroups according to their microbial configuration on principal coordinate projections, with PCo2 specifically shown here (dichotomized by median). *Bacteroides uniformis* (also median dichotomized) is shown as the highest-loaded organism on this axis (Supplementary Data 8). We adjusted for age, race, Bristol stool scale, antibiotics use, alcohol consumption, body mass index, smoking, menopausal hormone use, physical activity, and calorie intake. Values represent odds ratios and 95% confidence intervals associated with each unit increase in prudent dietary score and 5 g increase in dietary fiber. Two-sided p-interaction was assessed using a Wald test of the product term of diet and PCo2 or abundance of *B. uniformis*. **a** Greater adherence to a prudent diet was associated with a greater reduction in the risk of diverticulitis among individuals with high PCo2. **b** Consistent with this, higher fiber intake was also associated with a greater reduction in the risk of diverticulitis among individuals with high PCo2. **c** *B. uniformis* modulated the association between prudent diet and diverticulitis, with a stronger inverse association among individuals with high abundances of *B. uniformis*.

for the systematic analysis of microbial community-associated metabolomes for the prioritization of metabolites with potential bioactivity in a phenotype of interest. Metabolomic features altered in diverticulitis such as enriched 1,7-dimethyluric acid and depleted lipid markers such as ceramides, dihydrocholesterol, and 7alpha-hydroxy-4-cholesten-3-one as well as indolin-2-one were highly-prioritized (Supplementary Data 12). For example, a group of unknown metabolites that covaried with Cer(d18:1/17:0) were highly prioritized (Supplementary Fig. 7). Several of them were likely to be bacteria-produced ceramides based on m/z values, such as dihydroceramide C35 ($C_{35}H_{71}NO_4$, feature 14025, m/z = 570.5125) and 17:0(2R-OH) ceramide ($C_{35}H_{69}NO_4$, feature 13976, m/z = 568.4975)[50]. These unknown features corresponded to microbial profiles (Supplementary Fig. 8), including inversely correlated with *R. gnavus* and *Clostridium* pathogens and an enzyme involved in sphingolipid metabolism (EC 3.2.1.23, beta-galactosidase) but positively correlated with *Firmicutes bacterium*, *Subdoligranulum sp*, and 1,4-alpha-glucan branching enzyme (EC 2.4.1.18).

## Integrative multi-omic analysis reveals differential microbe-metabolite associations in diverticulitis

We detected a variety of microbe-metabolite correlation clusters across the whole samples (Supplementary Fig. 9). For example, a cluster of strong, positive associations between deoxycholic acid and *Blautia sp. CAG:257*, *F. plautii*, *R. gnavus*, and *C. bolteae* were observed, in line with the bile acid 7α-dehydroxylation activity potentially possessed by many of these microbes[55,56]. This group of microbes was also positively associated with tyramine. Clostridia genera including *Blautia*, *Clostridium*, and *Ruminococcus* have been documented to produce aromatic amines through the action of aromatic amino acid decarboxylase of tyrosine[57]. Consistent with the role of *Parasutterella* in bile acid and cholesterol metabolism[32], here, *P. excrementihominis* was negatively associated with cholic acid, alpha-muricholic acid, and carnitines. Finally, *Oscillibacter sp. 57_20*, which has been shown as among the indicators for a more favorable plant-based diet and cardiometabolic status[58], covaried with many other *Firmicutes bacterium*

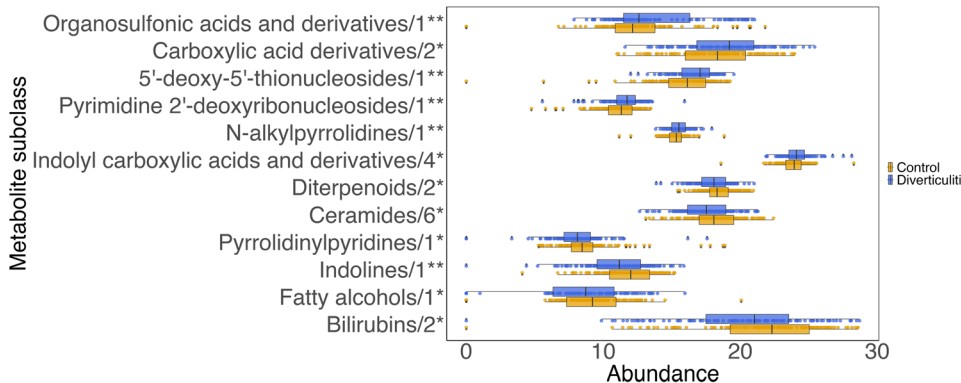

**Fig. 5 | Metabolomic shifts in diverticulitis. Metabolite subclasses (log2 transformed) that were significantly altered in diverticulitis.** Data are presented as median with the lower and upper hinges corresponding to the 25% and 75% percentiles. The lower and upper whiskers show the smallest or largest value within 1.5 IQR (inter-quartile range) from the nearest hinge. A total of 12 out of 84 subclasses were significantly associated with diverticulitis in a linear regression model (Supplementary Data 9). Subclasses were ranked by beta coefficients, with the number of metabolites in each group and two-sided, unadjusted $p$ value indicated (** $0.001 < p < 0.01$; * $0.01 < p < 0.05$).

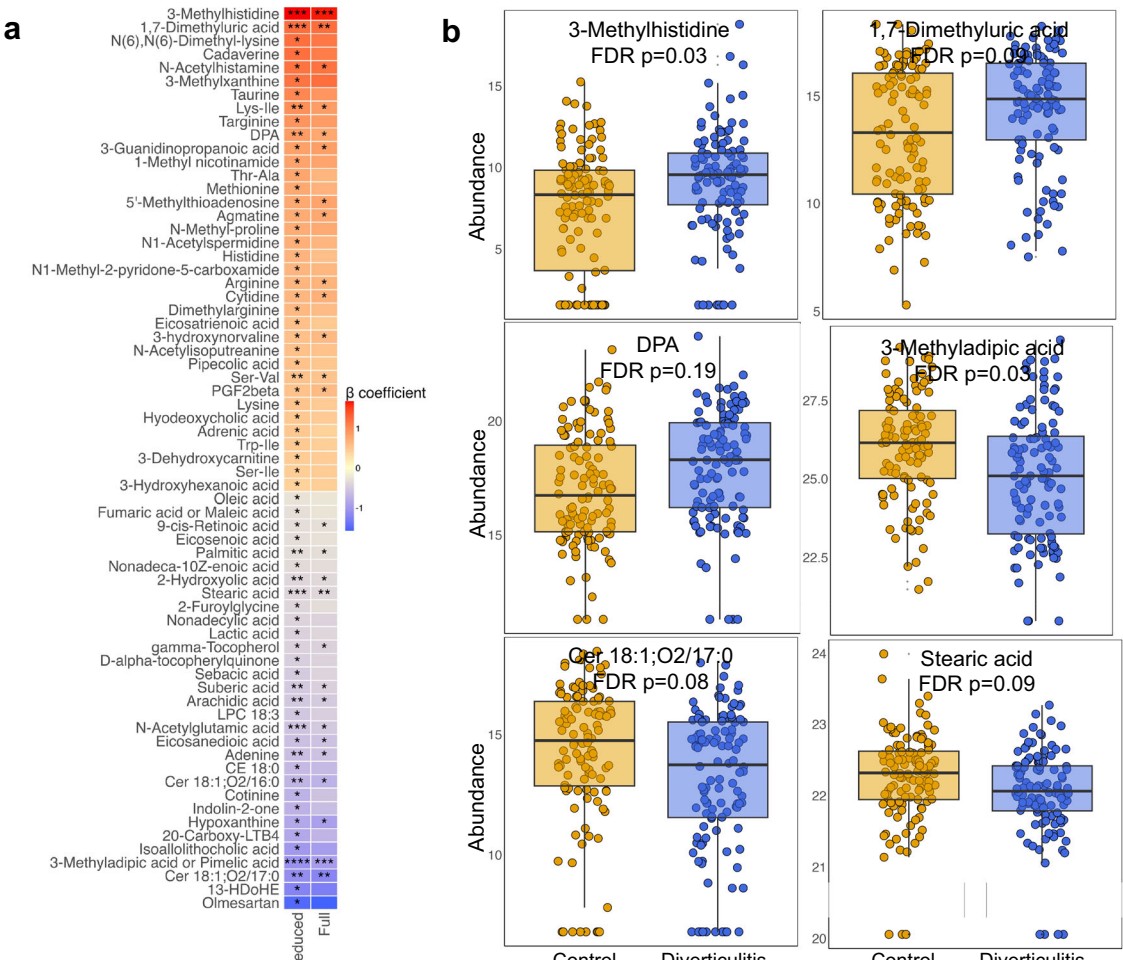

**Fig. 6 | Individual metabolites statistically significantly altered in diverticulitis. a** β coefficient of the statistically significant associations between metabolites and diverticulitis. We adjusted for age, race, Bristol stool scale, and antibiotics use in the reduced model, and the full model was further adjusted for fiber intake, alcohol consumption, body mass index, smoking, menopausal hormone use, physical activity, Alternate Healthy Eating Index, and calorie intake. *P* values are two-sided, with multiple comparison corrected using Benjamini-Hochberg false discovery rate (FDR). **** FDR $p < 0.01$; *** $0.01 < $ FDR $p < 0.05$; ** $0.05 < $ FDR $p < 0.1$; * $0.1 < $ FDR $p < 0.25$. Complete results for associations between metabolites and diverticulitis from MaAsLin 2 models are provided in Supplementary Data 10. **b** Examples of log2 transformed abundances of metabolites altered in diverticulitis. Data are presented as median with the lower and upper hinges corresponding to the 25% and 75% percentiles. The lower and upper whiskers show the smallest or largest value within 1.5 IQR (inter-quartile range) from the nearest hinge.

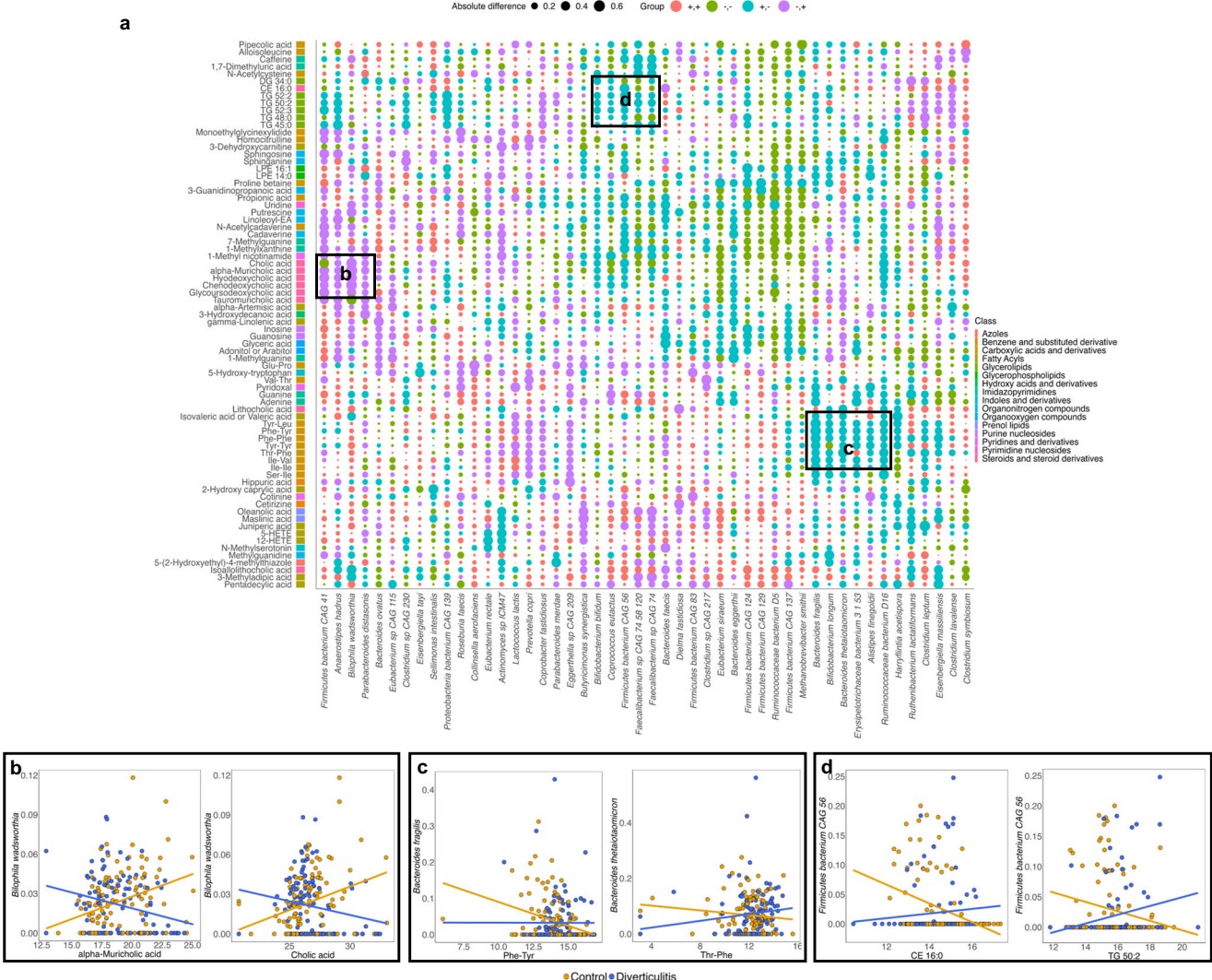

**Fig. 7 | Microbe-metabolite relationships differ in diverticulitis relative to control microbiomes. a** Differences in the Spearman correlations between microbial species and annotated metabolites in diverticulitis vs. controls. Only features with values >0.45 or <−0.45 (abs(diff) >0.45) are shown (data provided in Supplementary Data 13). Features are grouped by hierarchical clustering. The size of the dots represents the absolute value of the difference, and the color represents the directions of correlations in diverticulitis and controls, respectively. Amino acids, bile acids, triglycerides, and diet-derived metabolites were among those differentially correlated with specific species in diverticulitis and controls. **b** For example, the pathobiont *Bilophila wadsworthia*, known to expand in response to a Western diet resulting in intestinal inflammation and bile acid dysmetabolism[59,60], was positively associated with alpha-muricholic acid and cholic acid in controls but inversely associated in diverticulitis. **c** *Bacteroides fragilis* was inversely associated with amino acids in controls but positively associated in diverticulitis. **d** *Firmicutes bacterium CAG 56* was inversely associated with lipids such as CE 16:0 and TG 50:2 in controls but positively associated with diverticulitis.

and fiber-degrading bacteria by positively associating with plant-based derivatives such as sebacic acid (a metabolite of castor oil) but inversely associating with animal-based products (e.g., histidine-related metabolites, carnitines, and tyramine).

To further identify co-varying microbes and metabolites that were specifically relevant to diverticulitis, we calculated the differences in the correlations between each species and metabolite pair in diverticulitis vs. controls (Fig. 7a, Supplementary Data 13). This identified patterns of putative microbial chemistry that were differential (either newly-present or -absent) in diverticulitis. For example, *Bilophila wadsworthia* was positively correlated with several primary bile acids (e.g., cholic acid and alpha-muricholic acid) and secondary bile acids (e.g., hydrodeoxycholic acid) in controls. This is concordant with its known role as a bile acid metabolizer in typical Western populations, particularly in response to dietary components[59,60]. However, this relationship was lost in diverticulitis (where it was negatively correlated with bile acid levels; Fig. 7b). Likewise, *Bacteroides* members such as *B. fragilis* and *B. thetaiotaomicron* are involved in amino acid biosynthesis[61] and were inversely correlated with a group of amino acids in controls, which however, did not maintain or became positive in diverticulitis (Fig. 7c). *Firmicutes bacterium CAG 56* and *41*, which two covaried with many other *Firmicutes bacterium*, were inversely correlated with several lipid markers such as CE 16:0 and TG 50:2 in controls, but again, this relationship was disrupted in diverticulitis (Fig. 7d).

Many other microbe-metabolite modules besides the above-mentioned were differential in diverticulitis. These results, in combination with the individual perturbations of microbial composition, functions, and metabolomic features support microbe-metabolite interaction in diverticulitis and lay a foundation for understanding how chemical crosstalk drives and responds to inflammation in the condition, particularly with respect to dietary involvement where many of these chemical pools likely originate from.

## Discussion

Here, we report the largest and most comprehensive investigation of gut microbial perturbations in diverticulitis to-date, including enriched pro-inflammatory species such as *Ruminococcus gnavus* and opportunistic pathogens, as well as depleted butyrate producers such as *Eubacterium eligens* and *Subdoligranulum sp.* Enriched *Anaerotruncus colihominis* and *Acidaminococcus intestini* and depleted *Odoribacter splanchnicus* were found to be particularly associated with severe diverticulitis. Relatedly, highly prioritized metabolomic features associated with diverticulitis included enriched metabolites in histidine metabolism, 1,7-dimethyluric acid, and depleted lipid markers such as ceramides, dihydrocholesterol, and 7alpha-hydroxy-4-cholesten-3-one. Notably, we were able to relate many of these chemical perturbations with specific dysbiotic microbes, providing initial insight as to how microbial metabolism may be responding to or driving diverticulitis. Relatedly, the protective associations of prudent diet and dietary fiber with diverticulitis were modulated by microbial composition, including Bacteroidetes phylum members. Collectively, these data offer human evidence supporting the critical role of alterations in the gut microbiome composition and metabolic activity in diverticulitis.

Limited data suggest that microbial dysbiosis may play a key role in the pathogenesis of diverticulitis. However, interpretations of results from prior studies have been challenged by the small sample size, different sample sources and microbiome location (stool, mucosal biopsy, rectal swab), a lack of high-resolution characterization of the microbiome community, variable disease phenotyping, and the difficulty in identifying an appropriate control group[62,63]. For example, in surgically resected mucosal samples, higher occurrence of *Bifidobacterial longum* and counts of total *Bifidobacterium* were observed in patients with diverticulitis (*n* = 9) compared to those with colorectal cancer or IBD[20]. In another study of rectal swabs collected from 31 patients with uncomplicated acute diverticulitis compared to 25 controls with mixed gastrointestinal indications, *Proteobacteria* distinguished between diverticulitis and controls, and *Enterobacteriaceae* identified as the most discriminative taxa[21]. *Microbacteriaceae* and *Ascomycota* were enriched in the colonic segment chronically affected by diverticulitis compared to adjacent non-affected tissue[64]. When comparing severe diverticulitis to non-severe cases, we identified significant enrichment of *Anaerotruncus colihominis* and *Acidaminococcus intestini*, while also observing a decrease in *Odoribacter splanchnicus*. All three microbes are involved in sulfur metabolism. Our findings align with the Protolese et al. study, which also reported a marked increase in the abundance of sulfur-reducing microbes in the colonic tissues of patients with complicated diverticulitis compared to uncomplicated diverticulitis, and the complicated diseased segments demonstrated an increased sulfur-reducing and sulfur-oxidizing bacteria compared to non-diseased, adjacent normal regions[65]. Our results are also in line with a recent larger study comparing rectal swab samples from patients with acute diverticulitis (*n* = 65) and controls without diverticulitis or other gastrointestinal diseases, including reduced richness and Shannon alpha diversities and differences in the microbiome composition, such as reduced *Subdoligranulum*, *Faecalibacterium*, and *Parasutterella*[66]. However, interpreting the results from this study comparing complicated vs. uncomplicated diverticulitis proved to be challenging. Interestingly, all the genera that were found to be depleted in uncomplicated diverticulitis when compared to the control group, including *Anaerostipes*, *Bacteroides*, *Bifidobacterium*, *Faecalibacterium*, *Fusicatenibacter*, *Lachnospiraceae FCS020 group*, *Ruminiclostridium 5*, and *Subdoligranulum*, became instead enriched in complicated diverticulitis as compared to uncomplicated cases[66]. It should be noted that these results were from crude comparisons, neglecting to account for other factors that are related to both microbiome and diverticulitis, such as age, sex, obesity, and various lifestyle factors. In addition, patients with complicated diverticulitis might have been administered intravenous antibiotics which could rapidly change the mucosal-associated microbiome.

Similarly, prior studies have also examined metabolic changes in fecal and urinary samples of patients with diverticular disease, but evidence specifically pertaining to diverticulitis has been scant[22,23]. For example, patients with SUDD exhibited significantly lower levels of valerate, butyrate, and choline and higher levels of N-acetyl derivatives and an unknown metabolite[22]. In a pilot study, significant changes in 18 specific molecules such as tryptophan and phenylalanine were observed in SUDD patients undergoing treatment[67]. Moreover, six urinary metabolites showed high accuracy in discriminating diverticular disease from control subjects[23].

Many altered microbial members and metabolites we identified were inflammation-related. For example, *R. gnavus*, a prominent microbe in IBD producing an inflammatory polysaccharide[68], was also enriched in diverticulitis. *C. bolteae* and *C. clostridioforme* are opportunistic pathogens that have been isolated in intra-abdominal infections[26,27]. *Clostridium* species were significantly higher in autistic children compared to controls[69], and gastrointestinal symptoms that autistic patients often suffer from could be due to an overabundance of *C. bolteae* and an immunogenic polysaccharide product[70]. In contrast, *E. eligens* and *Faecalibacterium* produce butyrate and other short-chain fatty acids through fermentation of dietary fiber and exert anti-inflammatory effects[30]. *Subdoligranulum*, closely related to *Faecalibacterium*, is also known to produce butyrate[31] and found to be negatively associated with body weight, C-reactive protein (CRP), cirrhosis, and markedly reduced in IBD[38,71,72]. 3-methylhistidine, which was enriched in diverticulitis, is a marker for meat intake[73] and skeletal muscle loss[74] and has been related to frailty, inflammation, and DSS-induced colitis[53,75,76]. Similarly, as a metabolite of caffeine, the circulating level of 1,7-dimethyluric acid has also been associated with CRP[77]. We also found a reduction in ceramides in diverticulitis. While host-derived ceramides were generally found to be elevated in IBD[5], *Bacteroides*-derived sphingolipids were decreased in IBD which were correlated with intestinal inflammation and altered host ceramide pools[50]. Of note, Cer(d18:1/16:0) and Cer(d18:1/17:0) that were found depleted in diverticulitis were also among the lists of ceramides produced by *B. thetaiotaomicron* and *B. ovatus*[50]. These results taken together support a prolonged inflammatory response potentially mediated by gut microbiome composition and metabolites underlying the pathogenesis of diverticulitis.

The pronounced enriched functional potential in fatty acid biosynthesis was consistent with prior observations of obesity, in particular visceral fat, being associated with an increased risk of diverticulitis[78–80]. The significantly depleted and most predictive *Parasutterella* in our study was shown to increase in response to a resistant potato starch diet, and reductions in LDL cholesterol depended on higher levels of *Parasutterella*[81]. Colonization of murine *Parasutterella* led to changes in microbial-derived metabolites such as aromatic amino acid, bilirubin, purine, and bile acid derivatives, and the impacted bile acid profile supported the role of *Parasutterella* in cholesterol metabolism[32]. The metabolomics pathway that *Parasutterella* showed the strongest association with was fatty acid biosynthesis[33]. These lines of evidence indicate that the role of gut microbiota in diverticulitis may also be related to dysregulated lipid metabolism.

While previous studies have linked the gut microbiome to blood lipidemic, inflammatory, and glycemic markers in generally healthy populations[58], very limited data have connected microbiome composition and metabolic activity in the context of diseases. The many interesting covarying relationships observed between microbial features and metabolites and potentially distinct patterns according to diverticulitis provided a framework for integrating high-dimensional multi-omic data with biological interactions. For example, in animal models, *B. wadsworthia* expanded in response to a high-fat diet, which

was mediated by increased production of taurine-conjugated bile acids, leading to intestinal barrier dysfunction and inflammation[59,60]. Here, consistently, we saw an interplay between *B. wadsworthia* and a group of bile acids that might affect diverticulitis.

The finding on a diet-microbiome interaction on diverticulitis provides further support for the notion that gut microbiome modulates the effects of diet on chronic inflammation and inflammatory diseases. In a prior study of longitudinal samples from 307 men, we found that the protective association between dietary fiber and circulating marker for systemic inflammation was significantly stronger among participants without carriage of *P. copri*[42]. *Prevotella* has also been shown to determine changes in microbial composition and metabolic responses such as glucose metabolism after fiber interventions[48,49]. Gnotobiotic mice harboring distinct microbial communities demonstrated varied responses in epigenetic, transcriptional, and metabolomic levels after exposure to different types of fibers, confirming that gut microbiome is causally linked to different effects of fiber[44]. While the mechanisms underlying the interaction effects are beyond the goal of the study, our results suggest that microbes may have both direct effects on inflammation and inflammatory diseases such as diverticulitis, as well as indirect effects caused by altered availability of microbe-accessible dietary fiber or other fermentation products to other microbes. Further research is warranted to provide functional insights into the role of the gut microbiome in modulating dietary effects on inflammation and host health.

By leveraging a large-scale stool collection within a prospective cohort of health professionals spanning decades of follow-up, we were able to collect detailed information on lifestyle and dietary factors to minimize confounding for evaluating gut microbiome and metabolite alterations in diverticulitis, as well as to assess modulation effects of diet on diverticulitis by the gut microbiome. We used a previously validated instrument to ascertain diverticulitis that covered a full spectrum of disease severity. However, potential limitations should be noted. First, to facilitate large-scale microbiome analysis, we focused on stool microbiome, not mucosal. Studies have shown that the stool and mucosal microbial communities are clustered within individuals[82], and thus fecal samples can be used as a readily procured proxy for defining interpersonal differences in gut microbial ecology[83]. In addition, relevant biopsies are difficult to obtain since colonoscopy is usually avoided during the acute episode of diverticulitis[84]. Second, based upon the population-based design of our study, it was not feasible to access medical records for all cases of diverticulitis, which limited our ability to determine the exact timing of diagnosis for each individual. Thus, some cases might have been diagnosed before the time of stool collection. Future studies are warranted to examine the prospective role of the gut microbiome and metabolites in the development of diverticulitis. Third, we do not have accurate information on whether controls have diverticulosis or not. While prior colonoscopy-based studies have not identified a specific microbial signature in the mucosa of asymptomatic diverticulosis cases[85,86], we hypothesize that perturbations of gut microbiota composition and metabolic activity we observed may reflect changes relating to both formation and inflammation of diverticula. Finally, we examined the gut microbiome and metabolome in relation to diverticulitis only in women. The microbiome can vary between men and women[87], and there are certain female-specific risk factors for diverticulitis (e.g., menopausal hormone use)[88]. Further research involving men is necessary to validate our findings.

Our study offers evidence on alterations in gut microbiome composition, functions, and metabolic activity in diverticulitis, in particular those involved in inflammation and lipid metabolism. It proved to modulate dietary effects and corresponding metabolites relevant to diverticulitis. These results significantly expand our understanding of the pathogenesis of this understudied disease. Additional research, including validation among a broader range of

populations and well-designed prospective investigations, holds the potential in advancing clinical care by enabling personalized microbiome-informed dietary interventions or therapeutic manipulation of gut microbiota for the prevention and treatment of diverticulitis.

## Methods

### Study population and stool sample collection

Our study was approved by the Institutional Review Boards (IRBs) of the Brigham and Women's Hospital and the Harvard T.H. Chan School of Public Health. Written informed consent from participants have been obtained. We performed a nested case-control study of 121 patients with diverticulitis matched to 121 diverticulitis-free participants according to age, sex, and month of stool collection from the Nurses' Health Study (NHS) II. The NHS II is an ongoing prospective cohort of 116,429 female registered nurses residing across the U.S. aged 25–42 years at enrollment in 1989. Participants have been followed biennially by querying lifestyle, medical or other health-related information. Diet was assessed in 1991 and updated every four years thereafter using a validated semiquantitative food frequency questionnaire. The follow-up rate has been greater than 90% in most biennial cycles.

Beginning in February 2019, a large-scale prospective collection of fecal and oral microbiome samples was launched for NHS II participants, known as the Microbiome among Nurses (Micro-N) project. As of July 2021, a total of 14,992 women have returned a collection kit. Among these participants, we ascertained 121 incident diverticulitis requiring antibiotics or hospitalization based on information reported in the 2019 biennial questionnaire, including 19 cases considered as severe diverticulitis for requiring surgery or having an abscess. The validity of ascertaining diverticulitis in our cohort using this approach has been confirmed[89]. In a review of 226 medical records from NHS II participants reporting incident diverticulitis, self-report was confirmed in 92% of the cases with evidence from computed tomography examination, a surgical pathology report, or a provider diagnosis with a clinical presentation consistent with diverticulitis (e.g., abdominal pain, elevated white blood cells, fever).

As previously described in detail[25], stool samples were self-collected by participants using provided kits including three different sample tubes–95% ethanol, commercial OMNIgene-GUT kit, and a cryovial pre-filled anaerobically with liquid dental transport medium (Anaerobe Systems, CA, USA). Prior validation studies by our group and others showed that self-collected stool samples using these preservatives provided statistically near-identical multi-omic data to fresh frozen samples[25,90,91]. Participants were asked to collect samples from the same bowel movement for all three tubes. Participants were also asked to complete a brief stool sample questionnaire. The kits were returned via U.S. mail and stored at −80 °C freezers until nucleic acid extraction.

### Metagenomic profiling

We performed high-output microbial whole genome sequencing on OMNIgene-GUT samples at the Broad Institute (Cambridge, MA, USA). Stool DNA and RNA co-extractions were carried out using the AllPrep PowerFecal DNA/RNA Kit (QIAGEN, Inc.). DNA was quantified by Quant-iT PicoGreen dsDNA Assay (Life Technologies). Illumina sequencing libraries were prepared using the Nextera XT DNA Library Preparation kit (Illumina) according to the manufacturer's recommended protocol, with reaction volumes scaled accordingly. Batches of 184 libraries were pooled by transferring equal volumes of each uniquely barcoded library. Metagenomic libraries were sequenced on the Illumina NovaSeq 6000 platform, targeting ~3 Gb of sequence per sample with 151 bp, paired-end reads. One sample without reads was excluded from further analysis.

Taxonomic and functional profiles were generated by using the bioBakery 3 meta'omics workflow[92]. Sequencing reads were passed

through the KneadData 0.12.0 quality control pipeline (http://huttenhower.sph.harvard.edu/kneaddata) to remove low-quality read bases and reads of human origin. Taxonomic profiling was performed using MetaPhlAn 3.1.0 (https://huttenhower.sph.harvard.edu/metaphlan)[92], which classifies the metagenomic reads to taxonomies and yields relative abundances of taxa identified in the sample based on approximately 1.1 million clade-specific marker genes derived from 871,000 microbial genomes (corresponding to >13,500 bacterial and archaeal).

Metagenomes were functionally profiled using HUMAnN 3.5 (http://huttenhower.sph.harvard.edu/humann)[92]. Briefly, for each sample, taxonomic profiling is used to identify detectable organisms. Reads are recruited to sample-specific pangenomes including all gene families in any detected microorganisms using Bowtie2[93]. Unmapped reads are aligned against UniRef90[94] using DIAMOND translated search[95]. Hits are counted per gene family and normalized for length and alignment quality. For calculating abundances from reads that map to more than one reference sequence, search hits are weighted by significance (alignment quality, gene length and gene coverage). UniRef90 abundances from both the nucleotide and protein levels were then mapped to level 4 Enzyme Commission (EC) nomenclature and combined into structured pathways from MetaCyc[96].

### Metabolomic profiling

**LC–MS profiling.** A combination of four LC–MS methods was used to profile metabolites in the fecal homogenates, as previously published[97]; two methods that measure polar metabolites, a method that measures metabolites of intermediate polarity (for example, fatty acids and bile acids), and a lipid profiling method. The 95% ethanol solution in which stool samples were preserved was aliquoted into two 10 μL and two 30 μL aliquots in 1.5 mL centrifuge tubes for LC-MS sample preparation. Reference pooled material was generated by combining aliquots of each sample in the study. For the analysis queue in each method, subjects were randomized, and pairs of pooled reference samples were inserted into the queue at intervals of approximately 20 samples for quality control and data standardization. Samples were prepared for each method using extraction procedures that are matched for use with the chromatography conditions. Data were acquired using LC–MS systems comprised of Nexera X2 U-HPLC systems (Shimadzu Scientific Instruments) coupled to Q Exactive/Exactive Plus orbitrap mass spectrometers (Thermo Fisher Scientific). The method details are summarized below.

**Method 1: HILIC-pos (positive ion mode MS analyses of polar metabolites).** LC–MS samples were prepared from stool ethanol extracts (10 μl) with the addition of nine volumes of 74.9:24.9:0.2 v/v/v acetonitrile/methanol/formic acid containing stable isotope-labeled internal standards (valine-d8, Isotec; and phenylalanine-d8, Cambridge Isotope Laboratories). The samples were centrifuged (10 min, 9000 g, 4 °C), and the supernatants injected directly onto a 150 × 2-mm Atlantis HILIC column (Waters). The column was eluted isocratically at a flow rate of 250 μl/min with 5% mobile phase A (10 mM ammonium formate and 0.1% formic acid in water) for 1 min followed by a linear gradient to 40% mobile phase B (acetonitrile with 0.1% formic acid) over 10 min. MS analyses were carried out using electrospray ionization in the positive ion mode using full scan analysis over m/z 70–800 at 70,000 resolution and 3-Hz data acquisition rate. Additional MS settings are: ion spray voltage, 3.5 kV; capillary temperature, 350 °C; probe heater temperature, 300 °C; sheath gas, 40; auxiliary gas, 15; and S-lens RF level 40.

**Method 2: HILIC-neg (negative ion mode MS analysis of polar metabolites).** LC–MS samples were prepared from stool ethanol extracts (30 μl) with the addition of four volumes of 80% methanol containing inosine-15N4, thymine-d4 and glycocholate-d4 internal

standards (Cambridge Isotope Laboratories). The samples were centrifuged (10 min, 9000 g, 4 °C) and the supernatants were injected directly onto a 150 × 2.0-mm Luna NH2 column (Phenomenex). The column was eluted at a flow rate of 400 μl/min with initial conditions of 10% mobile phase A (20 mM ammonium acetate and 20 mM ammonium hydroxide in water) and 90% mobile phase B (10 mM ammonium hydroxide in 75:25 v/v acetonitrile/methanol) followed by a 10-min linear gradient to 100% mobile phase A. MS analyses were carried out using electrospray ionization in the negative ion mode using full scan analysis over m/z 60–750 at 70,000 resolution and 3 Hz data acquisition rate. Additional MS settings are: ion spray voltage, −3.0 kV; capillary temperature, 350 °C; probe heater temperature, 325 °C; sheath gas, 55; auxiliary gas, 10; and S-lens RF level 40.

**Method 3: C18-neg (negative ion mode analysis of metabolites of intermediate polarity; for example, bile acids and free fatty acids).** Stool ethanol extracts (30 μl) were mixed with 90 μl methanol containing PGE2-d4 as an internal standard (Cayman Chemical Co.) and centrifuged (10 min, 9000 g, 4 °C). The supernatants (10 μl) were injected onto a 150 × 2.1-mm ACQUITY BEH C18 column (Waters). The column was eluted isocratically at a flow rate of 450 μl/min with 20% mobile phase A (0.01% formic acid in water) for 3 min followed by a linear gradient to 100% mobile phase B (0.01% acetic acid in acetonitrile) over 12 min. MS analyses were carried out using electrospray ionization in the negative ion mode using full scan analysis over m/z 70–850 at 70,000 resolution and 3 Hz data acquisition rate. Additional MS settings are: ion spray voltage, −3.5 kV; capillary temperature, 320 °C; probe heater temperature, 300 °C; sheath gas, 45; auxiliary gas, 10; and S-lens RF level 60.

**Method 4: C8-pos.** LC–MS samples were prepared from stool ethanol extracts (10 μl) using 190 μl isopropanol containing 1-dodecanoyl-2-tridecanoyl-sn-glycero-3-phosphocholine as an internal standard (Avanti Polar Lipids; Alabaster, AL). After centrifugation (10 min, 9000 g, ambient temperature), supernatants (10 μl) were injected directly onto a 100 × 2.1-mm ACQUITY BEH C8 column (1.7 μm; Waters). The column was eluted at a flow rate of 450 μl/min isocratically for 1 min at 80% mobile phase A (95:5:0.1 v/v/vl 10 mM ammonium acetate/methanol/acetic acid), followed by a linear gradient to 80% mobile phase B (99.9:0.1 v/v methanol/acetic acid) over 2 min, a linear gradient to 100% mobile phase B over 7 min, and then 3 min at 100% mobile phase B. MS analyses were carried out using electrospray ionization in the positive ion mode using full scan analysis over m/z 200–1100 at 70,000 resolution and 3 Hz data acquisition rate. Additional MS settings are: ion spray voltage, 3.0 kV; capillary temperature, 300 °C; probe heater temperature, 300 °C; sheath gas, 50; auxiliary gas, 15; and S-lens RF level 60.

**Metabolomics data processing.** Raw LC–MS data were acquired to the data acquisition computer interfaced to each LC–MS system and then stored on a robust and redundant file storage system (Isilon Systems) accessed via the internal network at the Broad Institute. Nontargeted data were processed using Progenesis QIsoftware (v 2.0, Nonlinear Dynamics) to detect and de-isotope peaks, perform chromatographic retention time alignment, and integrate peak areas. To remove redundant adducts and features LC-MS data were clustered based on feature intensities and retention time. Clusters were generated containing features occurring within a retention time window of 0.025 minutes and with intensities correlating above Spearman rank correlation coefficients above 0.8. The feature with the highest mean abundance within each cluster was kept as the representative ion for each cluster and all other cluster members removed from the final dataset. Unclustered features (singlets) were kept in the final data set. Peaks of unknown identity were tracked by method, m/z and retention time. Identification of the resulting nontargeted metabolite LC–MS

peaks was conducted by: i) matching measured retention times and masses to mixtures of reference metabolites analyzed in each batch; and ii) matching an internal database of >1000 compounds that have been characterized using the Broad Institute methods. Temporal drift was monitored and normalized with the intensities of peaks measured in the pooled reference samples.

### Statistical analysis

**Sample size for microbiome and metabolomics analyses.** We generated taxonomic and functional profiles for 241 metagenomes and metabolomic profiles for 237 samples. After excluding samples with no reads and averaging data for five duplicate subjects, we included microbiome data from 235 participants (117 diverticulitis cases and 118 controls) and metabolomics data from 232 participants (116 diverticulitis cases and 116 controls) in the analyses. In the microbiome-metabolite interaction analysis, data from 113 diverticulitis cases and 113 controls were included.

**Overall microbiome and metabolite composition.** We calculated the Bray-Curtis dissimilarity to determine the differences in the overall composition of the microbiome and metabolite in diverticulitis and controls. We performed omnibus testing with permutational multivariate analysis of variance (PERMANOVA) to quantify the percentage of variance in the microbiome and metabolite profile explained by diverticulitis and other factors.

**Per-feature testing.** We excluded microbial species, pathways and ECs that did not surpass minimum prevalence (10% of samples) and relative abundance (0.01% for species and pathways and 0.001% for ECs) threshold. After filtering, a total of 189 microbial species, 377 pathways, 1744 enzymes, and 552 metabolites were included in the analyses. We identified individual features associated with diverticulitis using a multivariate linear model in MaAsLin 2[98]. We adjusted for age, race, Bristol stool scale, and antibiotics use in the reduced model, and the full model was further adjusted for fiber intake, alcohol consumption, BMI, smoking, menopausal hormone use, physical activity, Alternate Healthy Eating Index (AHEI), and calorie intake as Eq. 1:

$$\begin{aligned} \text{microbiome/metabolite features} \sim {} & \text{disease status} + \text{age} + \text{race} \\ & + \text{Bristol stool scale} + \text{antibiotics} + \text{fiber} + \text{alcohol} + \text{BMI} + \text{smoking} \\ & + \text{menopausal hormone} + \text{physical activity} + \text{AHEI} + \text{calorie} \end{aligned}$$

(1)

To identify microbiome/metabolite features associated with the severity of the disease, we also fitted MaAsLin 2 models comparing severe diverticulitis and controls to non-severe diverticulitis. We further included the much larger group of unannotated metabolites (*n* = 56,937 in total) in MACARRoN (https://huttenhower.sph.harvard.edu/macarron/) to prioritize novel, potentially bioactive metabolites based on guilt by association.

Overall type I error was controlled using the Benjamini-Hochberg false discovery rate. A corrected p-value of <0.25 was considered statistically significant.

**Random forest classification.** We used random forests as implemented in *scikit-learn* (1.0.2) package in Python to generate a microbiome-based classifier to discriminate between diverticulitis and controls. The model was fit with cross-validation with 100 random splits (ShuffleSplit in Python) and a 75/25 random split of training and testing folds. The performance of the RF classifier was quantified by calculating the AUC over 100 iterations. The top 20 features contributing to the classification were assessed via the Gini importance (mean decrease impurity) from the RF model.

**Derivation of prudent and Western dietary patterns.** We used the approach reported before to derive posteriori dietary patterns[15,99]. Briefly, we performed principal component analysis using the FACTOR Procedure in SAS based on 37 food groups. The obtained factors were rotated using orthogonal transformation (Varimax function in SAS) so that the factors were uncorrelated with each other and easier to interpret. We used the eigenvalue (>1), Scree test, and interpretability to determine the number of factors to retain. The factor score for each pattern was computed by combining the observed variables with weights that were proportional to their factor loadings. The first factor, the prudent dietary pattern, was characterized by high intake of vegetables, legumes, fruit, fish, and legumes. In contrast, the second factor, the Western dietary pattern, was characterized by a high intake of processed meat, red meat, French fries, sugary drinks, and refined grains.

**Association of diet and gut microbiome composition with diverticulitis.** We categorized participants into quartiles according to dietary patterns and intake of dietary fiber and evaluated their associations with diverticulitis using multivariate logistic regression models, with adjustment for age, race, alcohol consumption, BMI, menopausal hormone use, physical activity, and intakes of total calorie and red and processed meat. Models for prudent and Western dietary patterns were additionally adjusted for fiber. To test the interaction between diet and gut microbiome composition on diverticulitis, we included product terms of diet and pre-specified gut microbiome composition indicators (PCo1, PCo2, and relative abundances of *Bacteroides uniformis* and *Prevotella copri* as the highest loaded species on the axes) and assessed the significance of interaction using a Wald test.

**Microbe–metabolite interaction.** We used HAllA (Hierarchical All-against-All association, 0.8.20), which is a high-sensitivity pattern discovery in large, paired multi-omic datasets, to discover significant associations between metabolites and microbial features including species and functional potentials[100]. HAllA includes hierarchical clustering, all-against-all association testing between features, and statistical significance determination by permutation testing, and reports significant associations between clusters of related features ("block associations").

To further assess microbiome and metabolite interaction relevant to diverticulitis, we calculated the differences in the Spearman correlations between species and annotated metabolites in diverticulitis vs. controls. Features were grouped by hierarchical clustering of distance matrix. The output is the absolute value of the differences in the correlations as well as directions of the correlations in the diverticulitis and controls.

### Reporting summary

Further information on research design is available in the Nature Portfolio Reporting Summary linked to this article.

## Data availability

Based on the informed consent of the participants, the MassGeneral-Brigham IRB determined that individual-level data, including both phenotype and sequencing data, be provided to the scientific community through repositories with controlled access. Thus, metagenomic sequence data, metabolomics data, and phenotype data that support the findings of this study have been deposited in dbGaP (study accession: phs003531.v1.p1; BioProject: PRJNA1067170) with controlled access. Investigators can submit an online Data Access Request using NIH eRA Commons account to https://dbgap.ncbi.nlm.nih.gov/aa/wga.cgi?page=login, which will be forwarded to the investigator's designated Institutional Signing Official for approval and the NIDDK Central Repository Data Access Committee (niddk-dac@mail.nih.gov)

for review. In addition, investigators may contact Dr. Wenjie Ma at wma6@mgh.harvard.edu to facilitate access.

## Code availability

All codes have been deposited in GitHub repository (https://github.com/biobakery/MicroN-diverticulitis).

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

## Acknowledgements

This work is supported by the National Institutes of Health grants U01 CA176726, R01 DK101495 (L.L.S., A.T.C.), and K23 DK125838 (L.H.N.), the American Gastroenterological Association Research Foundation's Research Scholar Awards AGA2021-13-01 (WM), AGA2020-13-04 (L.H.N.), and the Crohn's and Colitis Foundation Career Development Award (L.H.N.). W.M. is supported by the MGH Claflin Distinguished Scholar Award. L.H.N. is the Chen Institute MGH Department of Medicine Transformative Scholar. C.H. is supported by R24 DK110499 from the National Institutes of Health. A.T.C. is an American Cancer Society Clinical Research Professor. The content is solely the responsibility of the authors and does not necessarily represent the official views of the funding agencies.

## Author contributions

Design and conduct of the study: W.M., L.L.S., C.H., A.T.C.; collection of data: W.M., L.H.N., M.S., C.B.C., L.L.S., C.H., A.T.C.; analysis of data: W.M., Y.W., L.H.N., R.S.M., J.H., A.B., L.J.M.; interpretation of data: all authors; preparation of the manuscript: W.M., L.L.S., C.H., A.T.C.; review and approval of the manuscript for submission for publication: all authors.

## Competing interests

A.T.C. served as a consultant for Pfizer Inc., Bayer Pharma AG, and Boehringer Ingelheim and received grants from Pfizer Inc., Zoe Ltd, and Freenome for work unrelated to the topic. Other authors have no conflicts of interest to disclose.
