## [Peer Review File · Nature Communications]

REVIEWER COMMENTS

Reviewer #1 (Remarks to the Author):

Comments for transmission to the authors

This large, prospective, cohort study assessed the gut microbiome and the metabolic activity in patients with history of diverticulitis.

Selecting 121 cases and 121 controls from a population of 14,992 women from the Nurses' Health Study II who returned a stool sample from February 2019 to July 2021 as part of the large-scale Microbiome among Nurses (Micro-N) study, authors found significant difference between cases and controls about gut microbiome composition and fecal metabolome.

This study is really interesting and timely, because the pathophysiology of diverticulitis is still unclear and microbiome imbalance is one of the pathogenetic hypothesis for its occurrence. However, this study suffers from several bias.

1. It is unclear which population has been investigated. Authors stated that "...we ascertained incident diverticulitis requiring antibiotics or hospitalization based on information reported in the 2019 biennial questionnaire. We also considered severe diverticulitis if the participant required surgery for diverticulitis or had an abscess. In a review of 226 medical records from NHS II participants reporting incident diverticulitis, self-report was confirmed in 92% of the cases....". However, it is unclear how many complicated (namely patients with abscesses or requiring surgery) and uncomplicated (if I have understood well, every other patient do not having with abscesses or not requiring surgery) occurred.

2. Author stated that "...In a review of 226 medical records from NHS II participants reporting incident diverticulitis, self-report was confirmed in 92% of the cases...". However, this statement is too general: how the diagnosis of diverticulitis was made?

3. Definition of diverticulitis has to be reported and referenced.

4. It is unclear when the stools were collected according to the incident diverticulitis: before? After? And how much time before or how many after? This is a key point of this study, because collection before or after diverticulitis may change the perspective about the role of microbiome imbalance.

5. There are no information about differences between patients taking antibiotics or not, and about the distance time from the antibiotic course (Months? Years?). Also this point is important, because we know that antibiotic treatment may influence significantly the intestinal microbiome imbalance for several months (Palleja A. Recovery of gut microbiota of healthy adults following antibiotic exposure. *Nat Microbiol* 2018;3(11):1255-1265).

6. Authors stated that *Fecalibacterium* species are significantly reduced in severe diverticulitis. How do the authors explain this finding, that differs from that recently reported by Portolese et al. (ref. 64) and O'Grady et al. (ref. 65)?

7. Authors cannot consider patients having abscesses and not requiring surgery and patients requiring surgery as part of the same group. This because the microbiome may differently works with or without surgery, in example influencing the occurrence of anastomotic healing (Hajjar R. Gut microbiota influence anastomotic healing in colorectal cancer surgery through modulation of mucosal proinflammatory cytokines. *Gut* 2023;72(6): 1143-1154). And this leads again to the point 5.

8. Authors assessed fecal microbiota only in females. This could be another bias, because microbiome of males and females may significantly differ (Mueller S. Differences in fecal microbiota in different European study populations in relation to age, gender, and country: a cross-sectional study. *Appl Environ Microbiol.* 2006 Feb;72(2):1027-33): this means that the results of this study cannot be generalized to overall population but just to the female population.

9. Metabolic changes have been also assessed in fecal and urinary samples of patients complaint for Symptomatic Uncomplicated Diverticular Disease (SUDD). In example, Tursi et al found that SUDD patients were characterized by significantly lower levels of valerate, butyrate, and choline, and by higher levels of N-acetyl-glucosamine, U1, and N-acetyl-glucosamine (ref.22). Moreover, preliminary study from the same group, and in the same population, found differences for 18 molecules (tryptophan, phenylalanine, tyrosine, 4-hydroxyphenylacetate, urocanate, X-6.363, X-5.779, uridylate, galactose, X-4.197, threonine, sarcosine, methionine, 2-oxoisocaproate, 5-aminolevulinate, alanine, leucine, valerate), which changed significantly under after treatment (Laghi A. Impact of treatments on fecal microbiota and fecal metabolome in symptomatic uncomplicated diverticular disease of the colon: a pilot study. *J Biol Regul Homeost Agents* 2018;32(5): 1421-1432). These data have to be cited and commented as comparison with what happens in diverticulitis.

Reviewer #2 (Remarks to the Author):

Important topic - diverticulitis is an extremely common and costly condition. In spite of this it's pathogenesis is poorly understood.

In the current study analysis of diet/lifestyle factors, microbiome and metabolomic profiling is a real strength. Several previous studies have shown alterations in microbiome. To show difference in the metabolome helps shed further light on the causality pathway. Furthermore it is an interesting finding to observe how microbiome composition mediates relative protective effect of fibre intake.

Temporality - it is unclear how the stool sample relates to the episode of diverticulitis, and indeed to treatment with antibiotics. Is this single stool sample representative of the individuals long term microbiome, or is it a consequence of either antibiotic use, or related to diverticulitis (if the stool sample is taken after diverticulitis), or other factor (eg bowel preparation)?

Do the controls have diverticulosis? Are the observed microbiome changes related to the presence of diverticulosis or diverticulitis? Without knowing if the controls have diverticulosis this is a limitation of the study.

Generalisability - women only. This should be listed as a limitation.

The conclusion should be clear that clinical implications / advances should still be in the setting of research / trials. The current evidence is not sufficient to introduce into routine clinical practice.

Reviewer #3 (Remarks to the Author):

General comment

The work of Ma et al. delves into the gut microbiology associated to diverticulitis. The authors analysed a cohort of 121 case/control pairs extracting condition-specific populations as well as connecting microbes, metabolites and diet in the light of diverticulitis.

The work is thorough and the paper well written, with very little that requires to be clarified further.

Major comments

1. Modelling: Was multi-collinearity taken into account for the random forest classifier or the interpretation of the top contributing features?

Minor comments

1. Line 89: Rephrase, not clear what “individually and mutually” means here (it is clear later but not at this point).
2. Table 1: Specify what the values in the table are (e.g., are they averages and standard deviations?)
3. Line 168: The process of filtering resulting in the 189 species should be mentioned here as well.
4. Figure 3: Specify what the area plots represent and label the axes.

REVIEWER COMMENTS

Reviewer #1 (Remarks to the Author):

Comments for transmission to the authors

This large, prospective, cohort study assessed the gut microbiome and the metabolic activity in patients with history of diverticulitis.

Selecting 121 cases and 121 controls from a population of 14,992 women from the Nurses' Health Study II who returned a stool sample from February 2019 to July 2021 as part of the large-scale Microbiome among Nurses (Micro-N) study, authors found significant difference between cases and controls about gut microbiome composition and fecal metabolome.

This study is really interesting and timely, because the pathophysiology of diverticulitis is still unclear and microbiome imbalance is one of the pathogenetic hypothesis for its occurrence. However, this study suffers from several bias.

1. It is unclear which population has been investigated. Authors stated that "...we ascertained incident diverticulitis requiring antibiotics or hospitalization based on information reported in the 2019 biennial questionnaire. We also considered severe diverticulitis if the participant required surgery for diverticulitis or had an abscess. In a review of 226 medical records from NHS II participants reporting incident diverticulitis, self-report was confirmed in 92% of the cases....". However, it is unclear how many complicated (namely patients with abscesses or requiring surgery) and uncomplicated (if I have understood well, every other patient do not having with abscesses or not requiring surgery) occurred.

Thank you for the clarification. Among the 121 diverticulitis cases, we had 19 cases classified as severe diverticulitis (i.e., requiring surgery or having an abscess). We have revised the text in the Methods as "Among these participants, we ascertained 121 incident diverticulitis requiring antibiotics or hospitalization based on information reported in the 2019 biennial questionnaire, including 19 cases considered as severe diverticulitis for requiring surgery or having an abscess (page 21, lines 512-515)."

2. Author stated that "...In a review of 226 medical records from NHS II participants reporting incident diverticulitis, self-report was confirmed in 92% of the cases...". However, this statement is too general: how the diagnosis of diverticulitis was made?

Thank you for the clarification. We have provided details on the diagnosis of diverticulitis and revised the sentence to the following, "The validity of ascertaining diverticulitis in our cohort using this approach has been confirmed¹. In a review of 226 medical records from NHS II participants reporting incident diverticulitis, self-report was confirmed in 92% of the cases with evidence from computed tomography examination, a surgical pathology report, or a provider diagnosis with a

clinical presentation consistent with diverticulitis (e.g., abdominal pain, elevated white blood cells, fever) (page 21, lines 515-520).”

3. Definition of diverticulitis has to be reported and referenced.

Thank you for the suggestion. As mentioned above to comment #2, we have reported the definition of diverticulitis with the reference (page 21, lines 515-520).

4. It is unclear when the stools were collected according to the incident diverticulitis: before? After? And how much time before or how many after? This is a key point of this study, because collection before or after diverticulitis may change the perspective about the role of microbiome imbalance.

We greatly appreciate you bringing up this excellent point. In our cohort, the identification of diverticulitis was established by participants through self-reports on questionnaires, which were administered biennially. It is important to note that based upon the population-based design of our study, it was not feasible to access medical records for all cases, which limits our ability to determine the exact timing of diagnosis for each individual. Nevertheless, a separate validation study among 226 participants reporting diverticulitis has demonstrated that the accuracy of identifying diverticulitis in our cohort using this method has been remarkably high.

Among the 117 stool microbiome samples from diverticulitis cases included in the analysis, 80 returned a baseline stool sample and then subsequently returned a questionnaire reporting diverticulitis during the preceding two-year interval. We believe this represents the most likely group in which the microbiome was characterized prior to diverticulitis.

In this instance, we demonstrate that the overall composition of the microbiome in cases who had the questionnaire returned after the stool collection (referred to as “incident”, n=80) was not significantly different from cases who had the questionnaire returned prior to stool collection (referred to as “prevalent”, n=37) (**Fig 1**). Using omnibus testing with PERMANOVA of Bray-Curtis dissimilarities, this timing indicator did not account for a significant amount of the variation in the microbiome profiles ($R^2=1.06\%$; $p=0.20$).

Figure 1: Overall microbiome composition in diverticulitis according to time of questionnaire return and stool collection. Principal coordinate analysis based on Bray-Curtis dissimilarities of species suggested the overall microbiome composition in diverticulitis who had the questionnaire returned after the stool collection (referred to as “incident”) was not systematically different from cases who had the questionnaire returned prior to stool collection (referred to as “prevalent”).

We further performed multivariable-adjusted analyses in MaAsLin 2 by only including the “incident” cases. Although some significant associations were attenuated probably because of reduced sample size and statistical power, the species significantly altered in diverticulitis were largely similar to those identified in the main analysis that included all cases (**Fig 2**).

Figure 2: Species abundances statistically significantly altered in diverticulitis. β coefficients of the statistically significant associations between species and diverticulitis in analyses using samples from all cases (All) and only incident cases (Incident). We adjusted for age, race, Bristol stool scale, antibiotics use, fiber intake, alcohol consumption, body mass index, smoking, menopausal hormone use, physical activity, Alternate Healthy Eating Index, and calorie intake. **** $p < 0.01$; *** $0.01 < p < 0.05$; ** $0.05 < p < 0.1$; * $0.1 < p < 0.25$.

Taken together, as we acknowledged in the Discussion (page 20, lines 479-482), we believe that future studies are warranted to examine the prospective role of the gut microbiome and metabolites in the development of diverticulitis. Meanwhile, we herein show that the timing does not appear to influence the interpretations of our findings.

5. There are no information about differences between patients taking antibiotics or not, and about the distance time from the antibiotic course (Months? Years?). Also this point is important, because we know that antibiotic treatment may influence significantly the intestinal microbiome imbalance for several months (Palleja A. Recovery of gut microbiota of healthy adults following antibiotic exposure. *Nat Microbiol* 2018;3(11):1255-1265).

Thank you for bringing up this matter. In our study, 12.4% of diverticulitis cases and 7.4% of controls reported antibiotic use in the past month. To mitigate any potential confounding caused by antibiotics, we adjusted for antibiotic use in our multivariate models. Additionally, we further

excluded participants who took antibiotics in the past month and found species significantly associated with diverticulitis were similar to those identified in the main analyses (**Fig 3**). Moreover, these signals demonstrated no significant associations with antibiotic use, providing further support that antibiotic use has minimal influence on our results.

Figure 3: Species abundances statistically significantly altered in diverticulitis. β coefficients of the statistically significant associations between species and diverticulitis in analyses using the whole study population (All) and excluding individuals who took antibiotics in the past month (Exclude_abx). The third column (Antibiotics) shows β coefficients for the associations between the species and antibiotic use. We adjusted for age, race, Bristol stool scale, fiber intake, alcohol consumption, body mass index, smoking, menopausal hormone use, physical activity, Alternate Healthy Eating Index, and calorie intake. **** $p < 0.01$; *** $0.01 < p < 0.05$; ** $0.05 < p < 0.1$; * $0.1 < p < 0.25$.

6. Authors stated that *Faecalibacterium* species are significantly reduced in severe diverticulitis. How do the authors explain this finding, that differs from that recently reported by Portolese et al. (ref. 64) and O’Grady et al. (ref. 65)?

We would like to clarify that in our analyses, we observed significant decreases in several *Faecalibacterium* species in diverticulitis overall. However, when comparing severe diverticulitis to non-severe cases, we identified significant enrichment of *Anaerotruncus colihominis* and

Acidaminococcus intestini, while also observing a decrease in *Odoribacter splanchnicus*. It is worth mentioning that all three microbes are involved in sulfur metabolism. In this sense, our findings align with the Protolese et al. study, which also reported an imbalance of sulfur-metabolizing bacteria in complicated diverticulitis. Nonetheless, our study was not directly comparable to the Protolese study, regarding the sample source and microbiome location (mucosal microbiome from surgically resected samples vs. fecal microbiome from stool samples), comparison of patient groups (complicated diverticulitis vs. recurrent uncomplicated diverticulitis), and other influencing factors (antibiotic use and bowel preparation prior to the surgery).

We also compared our results to the O'Grady et al. study which analyzed rectal swab samples from patients with acute diverticulitis and compared them to controls without diverticulitis or other gastrointestinal diseases. Our findings were in line with their observations, including the decreases in alpha diversity indexes and certain microbial members, such as *Subdoligranulum*, *Faecalibacterium*, and *Parasutterella*. However, interpreting the results from the O'Grady study comparing complicated vs. uncomplicated diverticulitis proved to be challenging. Interestingly, all the genera that were found to be depleted in uncomplicated diverticulitis when compared to the control group, including *Anaerostipes*, *Bacteroides*, *Bifidobacterium*, *Faecalibacterium*, *Fusicatenibacter*, *Lachnospiraceae FCS020 group*, *Ruminiclostridium 5*, and *Subdoligranulum*, became instead enriched in complicated diverticulitis as compared to uncomplicated cases. It is worth mentioning that all these results were from crude comparisons, neglecting to account for other factors that are related to both microbiome and diverticulitis, such as age, sex, obesity, and various lifestyle factors. In addition, patients with complicated diverticulitis might have been administered intravenous antibiotics which could rapidly change the mucosal-associated microbiome.

We have incorporated these comparisons into the Discussion (page 18, lines 383-404).

7. Authors cannot consider patients having abscesses and not requiring surgery and patients requiring surgery as part of the same group. This because the microbiome may differently works with or without surgery, in example influencing the occurrence of anastomotic healing (Hajjar R. Gut microbiota influence anastomotic healing in colorectal cancer surgery through modulation of mucosal proinflammatory cytokines. Gut 2023;72(6): 1143-1154). And this leads again to the point 5.

We appreciate this consideration. While evidence suggests that gut microbiota plays a role in influencing surgical colonic healing and postoperative outcomes, the impact of surgical procedures for diverticulitis on the gut microbiome, particularly in the long term, remains poorly understood. Among the 19 cases classified as severe diverticulitis, 4 required surgery, 8 had an abscess, and 7 had both abscess and surgery. The small number limited our ability to conduct multivariate analyses for subgroup analysis. However, we demonstrate that the overall composition of the microbiome in severe cases was not significantly different in those requiring surgery from those having abscess only (**Fig 4**). Using omnibus testing with PERMANOVA of Bray-Curtis dissimilarities, severity subgroup did not account for a significant amount of the variation in the microbiome profiles ($R^2=6.3\%$; $p=0.29$).

Figure 4: Overall microbiome composition in severe diverticulitis cases according to requiring surgery or not.

Additionally, the compositions of the 20 most abundant genera were similar between diverticulitis cases requiring surgery and those having abscess only (Fig 5).

Fig 5. Relative abundances of the top 20 genera in severe diverticulitis cases according to requiring surgery or not.

8. Authors assessed fecal microbiota only in females. This could be another bias, because microbiome of males and females may significantly differ (Mueller S. Differences in fecal microbiota in different European study populations in relation to age, gender, and country: a cross-sectional study. *Appl Environ Microbiol.* 2006 Feb;72(2):1027-33): this means that

the results of this study cannot be generalized to overall population but just to the female population.

We concur with this point. However, we would like to note that in the same study², there were no significant differences in major microbial communities between men and women. The functions of the gut microbiome relating to human health do not appear to substantially differ according to sex. We have acknowledged this as a potential limitation in the Discussion, “Finally, we examined the gut microbiome and metabolome in relation to diverticulitis only in women. The microbiome can vary between men and women³, and there are certain female-specific risk factors for diverticulitis (e.g., menopausal hormone use)⁴. Further research involving men is necessary to validate our findings (page 20, lines 486-489).”

9. Metabolic changes have been also assessed in fecal and urinary samples of patients complaint for Symptomatic Uncompliated Diverticular Disease (SUDD). In example, Tursi et al found that SUDD patients were characterized by significantly lower levels of valerate, butyrate, and choline, and by higher levels of N-acetyl–glucosamine, U1, and N-acetyl–glucosamine (ref.22). Moreover, preliminary study from the same group, and in the same population, found differences for 18 molecules (tryptophan, phenylalanine, tyrosine, 4-hydroxyphenylacetate, urocanate, X-6.363, X-5.779, uridylate, galactose, X-4.197, threonine, sarcosine, methionine, 2-oxoisocaproate, 5-aminolevulinate, alanine, leucine, valerate), which changed significantly under after treatment (Laghi A. Impact of treatments on fecal microbiota and fecal metabolome in symptomatic uncomplicated diverticular disease of the colon: a pilot study. J Biol Regul Homeost Agents 2018;32(5): 1421-1432). These data have to be cited and commented as comparison with what happens in diverticulitis.

Thank you for the suggestion. We have incorporated a detailed discussion regarding prior findings of metabolomic changes in diverticular disease, “Similarly, prior studies have also examined metabolic changes in fecal and urinary samples of patients with diverticular disease, but evidence specifically pertaining to diverticulitis has been scant^{5,6}. For example, patients with SUDD exhibited significantly lower levels of valerate, butyrate, and choline and higher levels of N-acetyl derivatives and an unknown metabolite⁵. In a pilot study, significant changes in 18 specific molecules such as tryptophan and phenylalanine were observed in SUDD patients undergoing treatment⁷. Moreover, six urinary metabolites showed high accuracy in discriminating diverticular disease from control subjects⁶ (page 19, lines 406-412).”

Reviewer #2 (Remarks to the Author):

Important topic - diverticulitis is an extremely common and costly condition. In spite of this its pathogenesis is poorly understood.

In the current study analysis of diet/lifestyle factors, microbiome and metabolomic profiling is a real strength. Several previous studies have shown alterations in microbiome. To show difference in the metabolome helps shed further light on the causality pathway. Furthermore it is an interesting finding to observe how microbiome composition mediates relative protective effect of fibre intake.

1. Temporality - it is unclear how the stool sample relates to the episode of diverticulitis, and indeed to treatment with antibiotics. Is this single stool sample representative of the individuals long term microbiome, or is it a consequence of either antibiotic use, or related to diverticulitis (if the stool sample is taken after diverticulitis), or other factor (eg bowel preparation)?

We appreciate you bringing up this point. As shown in prior studies^{8,9}, a single measurement of the fecal microbiome can provide long-term information regarding organismal composition and functional potential.

The comment on temporality corresponds to Review 1's comments # 4 and 5. Based upon the population-based design of our study, it was not feasible to access medical records for all cases, which limits our ability to determine the exact timing of diagnosis for each individual. Among the 117 diverticulitis cases included in the analysis, 80 returned a baseline stool sample and then subsequently returned a questionnaire reporting diverticulitis during the preceding two-year interval. We believe this represents the most likely group in which the microbiome was characterized prior to diverticulitis. To respond to the reviewer, we demonstrate that the overall composition of the microbiome in cases who had the questionnaire returned after the stool collection (referred to as "incident", n=80) was not significantly different from cases who had the questionnaire returned prior to stool collection (referred to as "prevalent", n=37) (**Fig 1**). Using omnibus testing with PERMANOVA of Bray-Curtis dissimilarities, this timing indicator did not account for a significant amount of the variation in the microbiome profiles ($R^2=1.06\%$; $p=0.20$).

We further performed multivariable-adjusted analyses in MaAsLin 2 by only including the "incident" cases. Although some significant associations were attenuated probably because of reduced sample size and statistical power, the species significantly altered in diverticulitis were largely similar to those identified in the main analysis that included all cases (**Fig 2**).

In our study, 12.4% of diverticulitis cases and 7.4% of controls reported antibiotic use in the past month. To mitigate any potential confounding caused by antibiotics, we adjusted for antibiotic use in our multivariate models. Moreover, we further excluded participants who took antibiotics in the past month and found species significantly associated with diverticulitis were similar to those identified in the main analyses. These signals demonstrated no significant associations with

antibiotic use, providing further support that antibiotic use has minimal influence on our results (Fig 3).

There was only a single control participant who underwent a colonoscopy and another control who underwent other bowel procedures in the past two months preceding the collection of stool samples. Additionally, gut microbiome composition and metabolome tended to recover to baseline in two weeks after bowel preparation. Therefore, our results are not expected to be influenced by bowel preparation.

Taken together, as we acknowledged in the Discussion, we believe that future studies are warranted to examine the prospective role of the gut microbiome and metabolites in the development of diverticulitis (page 20, lines 479-482). Meanwhile, we herein show that the timing of antibiotic use does not appear to influence the interpretations of our findings.

2. Do the controls have diverticulosis? Are the observed microbiome changes related to the presence of diverticulosis or diverticulitis? Without knowing if the controls have diverticulosis this is a limitation of the study.

The controls in our study were selected among women who did not have a diagnosis of diverticulitis, regardless of the presence of diverticulosis. We have collected data on self-reported diverticulosis from the questionnaire, similar to diverticulitis. However, it is important to note that the accuracy of diverticulosis reporting is limited as individuals may only become aware of a diagnosis after a colonoscopy or abdominal imaging, and if a healthcare provider documents diverticulosis and informs the individual. Among women in this age group, the estimated prevalence of diverticulosis exceeds 30%, indicating that a significant proportion of controls also have diverticulosis. Prior colonoscopy-based studies have not identified a specific microbial signature in the mucosa of asymptomatic diverticulosis cases^{10,11}. It should also be noted that unlike prior studies that have utilized inappropriate control group such as patients with other gastrointestinal disorders, our controls are derived from the same source population as our cases, which is more representative of the actual population at risk for developing diverticulitis. We hypothesize that perturbations of gut microbiota composition and metabolic activity we observed may reflect changes relating to both the formation and inflammation of the diverticula.

We have acknowledged this as a potential limitation in the Discussion, “Third, we do not have accurate information on whether controls have diverticulosis or not. While prior colonoscopy-based studies have not identified a specific microbial signature in the mucosa of asymptomatic diverticulosis cases^{10,11}, we hypothesize that perturbations of gut microbiota composition and metabolic activity we observed may reflect changes relating to both formation and inflammation of diverticula (page 20, lines 482-486).”

3. Generalisability - women only. This should be listed as a limitation.

We concur with this point. However, we would like to note that medication that are differentially prescribed in men and women, such as oral contraceptives, are likely to account for the variations

in microbial communities³. After accounting for medication use and other factors, the functions of the gut microbiome relating to human health do not appear to substantially differ according to sex. We have acknowledged this as a potential limitation in the Discussion, “Finally, we examined the gut microbiome and metabolome in relation to diverticulitis only in women. The microbiome can vary between men and women³, and there are certain female-specific risk factors for diverticulitis (e.g., menopausal hormone use)⁴. Further research involving men is necessary to validate our findings (page 20, lines 486-489).”

4. The conclusion should be clear that clinical implications / advances should still be in the setting of research / trials. The current evidence is not sufficient to introduce into routine clinical practice.

Indeed, further evidence is required to support the current discoveries. This validation should be conducted among a broader range of populations and through prospective studies. In the Conclusion, we have toned down our statement, “These results significantly expand our understanding of the pathogenesis of this understudied disease. Additional research, including validation among a broader range of populations and prospective investigations, holds the potential in advancing clinical care by enabling personalized microbiome-informed dietary interventions or therapeutic manipulation of gut microbiota for the prevention and treatment of diverticulitis (page 20, lines 495-498).”

Reviewer #3 (Remarks to the Author):

General comment

The work of Ma et al. delves into the gut microbiology associated to diverticulitis. The authors analysed a cohort of 121 case/control pairs extracting condition-specific populations as well as connecting microbes, metabolites and diet in the light of diverticulitis.

The work is thorough and the paper well written, with very little that requires to be clarified further.

Major comments

1. Modelling: Was multi-collinearity taken into account for the random forest classifier or the interpretation of the top contributing features?

Thank you for the clarification. Multi-collinearity does not influence the prediction accuracy of random forest models. However, multi-collinearity does affect how we interpret the importance of the top contributing features, as the estimates of importance are spread out among correlated features. We ranked the 189 microbial species based on their mean abundances and used a selective method to eliminate highly correlated features (any feature with an absolute value of Spearman correlation coefficient greater than 0.7 with previous features was removed), resulting in 182 species remaining. Interestingly, none of the top contributing features were eliminated during this process. This indicates that multi-collinearity had minimal effect on the importance of key contributing features in our analysis.

Minor comments

1. Line 89: Rephrase, not clear what “individually and mutually” means here (it is clear later but not at this point).

Thank you for the suggestions. We have revised the sentence, “We successfully identified distinct microbial species, enzymes, and metabolites that undergo specific changes in diverticulitis, as well as any co-occurring microbe-metabolite associations relevant to the disease (page 3, lines 60-62).”

2. Table 1: Specify what the values in the table are (e.g., are they averages and standard deviations?)

Values are means(SD) for continuous variables or percentages for categorical variables. We have included this in the footnote of Table 1 (page 4, line 77).

3. Line 168: The process of filtering resulting in the 189 species should be mentioned here as well.

Thank you for the suggestion. Since we also included the post-filtered 189 species in the per-feature testing in MaAsLin 2, we added the following text in the Results before we described

results in Fig 2, “A total of 189 microbial species remained in the analysis with the minimum prevalence (10% of samples) and relative abundance (0.01%) threshold (page 6, lines 104-105).”

4. Figure 3: Specify what the area plots represent and label the axes.

Figure 3 shows the density of the selected metagenomic pathways and functional potentials, with the x-axis as the relative abundances and the y-axis as the density. We have labeled the axes in the figure (page 9).

References

- 1 Gunby, S. A. *et al.* Smoking and Alcohol Consumption and Risk of Incident Diverticulitis in Women. *Clin Gastroenterol Hepatol*, doi:10.1016/j.cgh.2023.11.036 (2023).
- 2 Mueller, S. *et al.* Differences in fecal microbiota in different European study populations in relation to age, gender, and country: a cross-sectional study. *Appl Environ Microbiol* **72**, 1027-1033, doi:10.1128/AEM.72.2.1027-1033.2006 (2006).
- 3 Sinha, T. *et al.* Analysis of 1135 gut metagenomes identifies sex-specific resistome profiles. *Gut microbes* **10**, 358-366, doi:10.1080/19490976.2018.1528822 (2019).
- 4 Jovani, M. *et al.* Menopausal Hormone Therapy and Risk of Diverticulitis. *Am J Gastroenterol* **114**, 315-321, doi:10.14309/ajg.0000000000000054 (2019).
- 5 Tursi, A. *et al.* Assessment of Fecal Microbiota and Fecal Metabolome in Symptomatic Uncomplicated Diverticular Disease of the Colon. *J Clin Gastroenterol* **50 Suppl 1**, S9-S12, doi:10.1097/MCG.0000000000000626 (2016).
- 6 Barbara, G. *et al.* Gut microbiota, metabolome and immune signatures in patients with uncomplicated diverticular disease. *Gut* **66**, 1252-1261, doi:10.1136/gutjnl-2016-312377 (2017).
- 7 Laghi, L. *et al.* Impact of treatments on fecal microbiota and fecal metabolome in symptomatic uncomplicated diverticular disease of the colon: a pilot study. *J Biol Regul Homeost Agents* **32**, 1421-1432 (2018).
- 8 Faith, J. J. *et al.* The long-term stability of the human gut microbiota. *Science* **341**, 1237439, doi:10.1126/science.1237439 (2013).
- 9 Mehta, R. S. *et al.* Stability of the human faecal microbiome in a cohort of adult men. *Nature microbiology* **3**, 347-355, doi:10.1038/s41564-017-0096-0 (2018).
- 10 Jones, R. B. *et al.* An Aberrant Microbiota is not Strongly Associated with Incidental Colonic Diverticulosis. *Sci Rep* **8**, 4951, doi:10.1038/s41598-018-23023-z (2018).
- 11 van Rossen, T. M. *et al.* Microbiota composition and mucosal immunity in patients with asymptomatic diverticulosis and controls. *PLoS One* **16**, e0256657, doi:10.1371/journal.pone.0256657 (2021).

REVIEWERS' COMMENTS

Reviewer #1 (Remarks to the Author):

The authors have replied to all questions, and have to be congratulated for their effort. Almost all concerns have been discussed and improved, However, a point remain unclear.

It is still unclear when the stools were collected according to the incident diverticulitis, and therefore how this could influence the results reported by the authors. They have tried to give a detailed answer: "...Among the 117 stool microbiome samples from diverticulitis cases included in the analysis, 80 returned a baseline stool sample and then subsequently returned a questionnaire reporting diverticulitis during the preceding two-year interval. We believe this represents the most likely group in which the microbiome was characterized prior to diverticulitis. In this instance, we demonstrate that the overall composition of the microbiome in cases who had the questionnaire returned after the stool collection (referred to as "incident", n=80) was not significantly different from cases who had the questionnaire returned prior to stool collection (referred to as "prevalent", n=37) (Fig 1). Using omnibus testing with PERMANOVA of Bray-Curtis dissimilarities, this timing indicator did not account for a significant amount of the variation in the microbiome profiles ($R^2=1.06\%$; $p=0.20$)...". Again, I appreciate the effort of the authors to clarify this key point but, to me, it is still unclear. It almost seems that fecal microbiome before and after diverticulitis is the same, without any significant changes. And this is difficult to understand: really therapies for diverticulitis, dietary advices after diverticulitis, potential comorbidities and/or concomitant treatments occurring in the following two years have not influenced the microbiome? In my opinion, this is consequence of the retrospective design of the study and, to clarify definitely this, a comparison with a control group of patients with asymptomatic diverticulosis is mandatory. But also this comparison is not possible due to retrospective design (even if the authors even if the authors tried to justify this with a series of epidemiological data which however do not resolve the basic problem, see the response to referee #2).

My advice is to report clearly in the limitation of the study that its retrospective design does not permit to resolve definitely all questions raised during the analysis of the microbiome of these patients. Further prospective and well designed studies are therefore mandatory on this topic.

Reviewer #2 (Remarks to the Author):

Thank you for considering the reviewer comments and making changes as appropriate.

REVIEWER COMMENTS

Reviewer #1 (Remarks to the Author):

The authors have replied to all questions, and have to be congratulated for their effort. Almost all concerns have been discussed and improved, However, a point remain unclear. It is still unclear when the stools were collected according to the incident diverticulitis, and therefore how this could influence the results reported by the authors. They have tried to give a detailed answer: "...Among the 117 stool microbiome samples from diverticulitis cases included in the analysis, 80 returned a baseline stool sample and then subsequently returned a questionnaire reporting diverticulitis during the preceding two-year interval. We believe this represents the most likely group in which the microbiome was characterized prior to diverticulitis. In this instance, we demonstrate that the overall composition of the microbiome in cases who had the questionnaire returned after the stool collection (referred to as "incident", n=80) was not significantly different from cases who had the questionnaire returned prior to stool collection (referred to as "prevalent", n=37) (Fig 1). Using omnibus testing with PERMANOVA of Bray-Curtis dissimilarities, this timing indicator did not account for a significant amount of the variation in the microbiome profiles ($R^2=1.06\%$; $p=0.20$)....". Again, I appreciate the effort of the authors to clarify this key point but, to me, it is still unclear. It almost seems that fecal microbiome before and after diverticulitis is the same, without any significant changes. And this is difficult to understand: really therapies for diverticulitis, dietary advices after diverticulitis, potential comorbidities and/or concomitant treatments occurring in the following two years have not influenced the microbiome? In my opinion, this is consequence of the retrospective design of the study and, to clarify definitely this, a comparison with a control group of patients with asymptomatic diverticulosis is mandatory. But also this comparison is not possible due to retrospective design (even if the authors even if the authors tried to justify this with a series of epidemiological data which however do not resolve the basic problem, see the response to referee #2).

My advice is to report clearly in the limitation of the study that its retrospective design does not permit to resolve definitely all questions raised during the analysis of the microbiome of these patients. Further prospective and well designed studies are therefore mandatory on this topic.

Thank you for the suggestion. We have clearly acknowledged the limitation in the Discussion, "Second, based upon the population-based design of our study, it was not feasible to access medical records for all cases of diverticulitis, which limited our ability to determine the exact timing of diagnosis for each individual. Thus, some cases might have been diagnosed before the time of stool collection. Future studies are warranted to examine the prospective role of the gut microbiome and metabolites in the development of diverticulitis." In the final conclusion paragraph, we proposed "Additional research, including validation among a broader range of populations and well-designed prospective investigations, holds the potential in advancing clinical care..."